# FUNCTION VECTORS IN LARGE LANGUAGE MODELS

**Eric Todd,**[*] **Millicent L. Li, Arnab Sen Sharma, Aaron Mueller,**
**Byron C. Wallace, and David Bau**
Khoury College of Computer Sciences, Northeastern University

## ABSTRACT

We report the presence of a simple neural mechanism that represents an input-output function as a vector within autoregressive transformer language models (LMs). Using causal mediation analysis on a diverse range of in-context-learning (ICL) tasks, we find that a small number attention heads transport a compact representation of the demonstrated task, which we call a *function vector* (FV). FVs are robust to changes in context, i.e., they trigger execution of the task on inputs such as zero-shot and natural text settings that do not resemble the ICL contexts from which they are collected. We test FVs across a range of tasks, models, and layers and find strong causal effects across settings in middle layers. We investigate the internal structure of FVs and find while that they often contain information that encodes the output space of the function, this information alone is not sufficient to reconstruct an FV. Finally, we test semantic vector composition in FVs, and find that to some extent they can be summed to create vectors that trigger new complex tasks. Our findings show that compact, causal internal vector representations of function abstractions can be explicitly extracted from LLMs.

## 1  INTRODUCTION

Since the study of the lambda calculus (Church, 1936), computer scientists have understood that the ability for a program to carry references to its own functions is a powerful idiom. Function references can be helpful in many settings, allowing expression of complex control flow through deferred invocations (Sussman, 1975), and enabling flexible mappings from inputs to a target task. In this paper we report evidence that autoregressive transformers trained on large corpora of natural text develop a rudimentary form of function references.

Our results begin with an examination of in-context learning (ICL; Brown et al., 2020). ICL mechanisms have previously been studied from the perspective of making copies (Olsson et al., 2022) and from a theoretical viewpoint (Von Oswald et al., 2023; Garg et al., 2022; Dai et al., 2023), but the computations done by large models to generalize and execute complex ICL functions are not yet fully understood. We characterize a key mechanism of ICL execution: *function vectors* (FVs), which are compact vector representations of input-output tasks that can be found within the transformer hidden states during ICL. An FV does not directly perform a task, but rather it triggers the execution of a specific procedure by the language model (Figure 1).

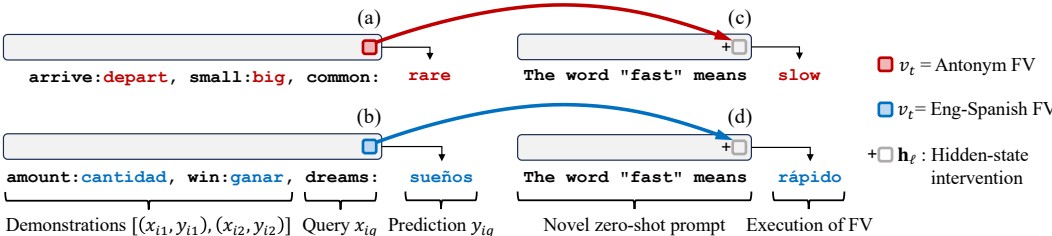

Figure 1: An overview of **function vectors** (FVs). An FV is extracted from activations induced by in-context examples of (a) antonym generation or (b) English to Spanish translation, and then inserted into an unrelated context to induce generation of (c) a new antonym or (d) translation.

[*]Correspondence to todd.er@northeastern.edu. Open-source code and data available at functions.baulab.info.

Function vectors arise naturally when applying causal mediation analysis (Pearl, 2001; Vig et al., 2020; Meng et al., 2022; 2023; Wang et al., 2022a) to identify the flow of information during ICL. We describe an activation patching procedure to determine the presence of a handful of attention heads that mediate many ICL tasks. These heads work together to transport a function vector that describes the task; the FV can be formed by summing outputs of the causal attention heads.

We test the hypothesis that function vectors are a general mechanism spanning many types of functions. To quantify the role and efficacy of function vectors, we curate a data set of over 40 diverse ICL tasks of varying complexity. We calculate FVs for these tasks and investigate impact of FVs in triggering those functions across a variety of LMs scaling up from 6B to 70B parameters.

We further ask whether FVs are portable: are the effects of an FV limited to contexts very similar to those where it is extracted, or can an FV apply in diverse settings? We compare the effects of FVs when inserted into diverse input contexts including differently-formatted forms, zero-shot formats, and natural text contexts. We find that FVs are remarkably robust, typically triggering function execution even in contexts that bear no resemblance to the original ICL context.

A key question is whether the action of FVs can be explained by word-embedding vector arithmetic (Mikolov et al., 2013; Levy & Goldberg, 2014; Merullo et al., 2023). We examine decodings of FVs (Nostalgebraist, 2020), and find that although FVs often encode a function's output vocabulary, those vocabularies do not fully identify an FV. In other words, to invoke functions, FVs need to carry some additional information beyond their encoding of the top vocabulary words.

Finally, we investigate whether the space of FVs has its own vector algebra over functions rather than words. We construct a set of composable ICL tasks, and we test the ability of FVs to obey vector algebra compositions. Our findings reveal that, to some extent, vector compositions of FVs produce new FVs that can execute complex tasks that combine constituent tasks. We emphasize that FV vector algebra is distinct from semantic vector algebra over word embeddings: for example, composed FV vectors can specify nonlinear tasks such as calculating the antonym of a word, that cannot themselves be implemented as a simple embedding-vector offset (Appendices A, L).

## 2 METHOD

### 2.1 A MOTIVATING OBSERVATION

When a transformer processes an ICL prompt with exemplars demonstrating task $t$, do any hidden states encode the task itself? We seek causal features rather than just correlations.

We can investigate this question with the following simple test: Gather a set of ICL prompts $P_t$ for the task $t$ and compute the average activation $\bar{\mathbf{h}}_\ell^t$ at the last token of each prompt at a particular layer $\ell$ of the model (Figure 2a). Then perform an intervention where $\bar{\mathbf{h}}_\ell^t$ is added to the representation after $\ell$ when the transformer completes a previously unseen zero-shot prompt (Figure 2b).

Surprisingly, we find that adding the average activations in this way at particular layers induces the model to perform the task in the new context. For example, if $t = antonym$, the red line in Figure 2c shows that adding $\bar{\mathbf{h}}_{12}^t$ at layer 12 in GPT-J causes the model to produce antonyms in a zero-shot context, with $24.3\%$ accuracy. That suggests that $\bar{\mathbf{h}}_{12}^t$ does encode the antonym task.

The effect of $\bar{\mathbf{h}}_\ell^t$ leads us to ask: Can we distill a more effective hidden-state representation of the task $t$? In the rest of Section 2 we describe an analysis of the mechanisms of ICL that leads to a function vector representation $v_t$ whose stronger causal effects are shown as a green line in Figure 2c.

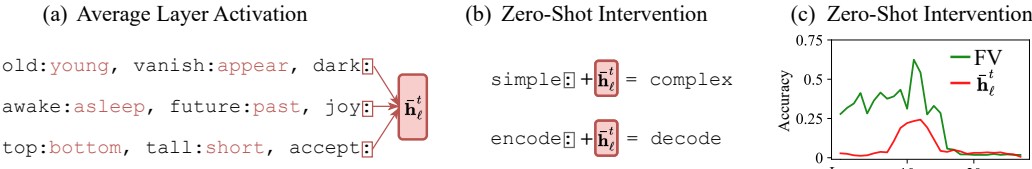

Figure 2: A motivating observation: (a) an average activation is computed over a set of antonym ICL prompts, and (b) added to a zero-shot context, which produces the opposite of unseen words. (c) Systematic effects (in red) for adding $\bar{\mathbf{h}}_\ell^t$ in middle layers of the network; even stronger effects are seen by the FV (in green).

## 2.2 FORMULATION

An autoregressive transformer language model $f$ takes an input prompt $p$ and outputs a next-token distribution $f(p)$ over vocabulary $\mathcal{V}$; we write $f(p)[y] \in [0, 1]$ for the predicted probability of output $y \in \mathcal{V}$ in response to input $p$. Internally, $f$ comprises $L$ layers; we examine their calculations at the last token position. Each layer $\ell \leq L$ has a vector representation of the last token, $\mathbf{h}_\ell \in \mathbb{R}^d$, that is computed from the previous layer as $\mathbf{h}_\ell = \mathbf{h}_{\ell-1} + m_\ell + \sum_{j \leq J} a_{\ell j}$, where $m_\ell$ is the output of a multilayer perceptron, and $a_{\ell j}$ is the projection of the output of the $j$th attention head (out of $J$ heads) into the hidden state at layer $\ell$. This definition of $a_{\ell j} \in \mathbb{R}^d$ adopts the framing of Elhage et al. (2021) rather than that of Vaswani et al. (2017) (see Appendix B for details). Attention heads and hidden states can be viewed as functions of transformer input, so we shall write $a_{\ell j}(p)$ or $\mathbf{h}_\ell(p)$ to denote their values when the transformer processes input $p$. The transformer's decoder $D$ maps the last layer hidden state to the output distribution $D(\mathbf{h}_L(p)) = f(p)$.

For each task $t \in \mathcal{T}$ in our universe of ICL tasks $\mathcal{T}$ we have a data set $P_t$ of in-context prompts $p_i^t \in P_t$. Each prompt $p_i^t$ is a sequence of tokens with $N$ input-output exemplar pairs $(x, y)$ that demonstrate the same underlying task $t$ mapping between $x$ and $y$, and one *query* input $x_{iq}$ corresponding to a target (correct) response $y_{iq}$ that is not part of the prompt, that should be predicted by the LM if it generalizes correctly. We focus our analysis on successful ICL by including in $P_t$ only prompts $p_i^t$ where the prediction $f(p_i^t)$ ranks the correct answer $y_{iq}$ highest. We write one ICL prompt as

$$p_i^t = [(x_{i1}, y_{i1}), \cdots, (x_{iN}, y_{iN}), x_{iq}] \tag{1}$$

We also make use of uninformative ICL prompts $\tilde{p}_i^t \in \tilde{P}_t$ for which the labels are shuffled; we use the tilde to indicate a shuffled prompt $\tilde{p}_i^t = [(x_{i1}, \tilde{y}_{i1}), \cdots, (x_{iN}, \tilde{y}_{iN}), x_{iq}]$ in which there is no systematic relationship between any of the $x_{ik}$ and $\tilde{y}_{ik}$.

## 2.3 CAUSAL MEDIATION TO EXTRACT FUNCTION VECTORS FROM ATTENTION HEADS

To distill the information flow during ICL, we apply causal mediation analysis.

Given a transformer model $f$ and an ICL prompt $p_i^t \in P_t$ from a dataset representing task $t$, we prompt the model with only input-output pairs $(x_i, y_i)$. Therefore, the LM must infer the implicit relationship between these $(x, y)$ pairs to correctly predict the answer given a novel query $x_{iq}$. We seek to identify model components with a causal role in the prediction of $y_{iq}$. We restrict our analysis to the attention heads since those are the components used by transformer LMs to move information between different token positions (Vaswani et al., 2017; Elhage et al., 2021). Formally, for each attention head $a_{\ell j}$ and task dataset $P_t$, we take the mean of task-conditioned activations $\bar{a}_{\ell j}^t$ as

$$\bar{a}_{\ell j}^t = \frac{1}{|P_t|} \sum_{p_i^t \in P_t} a_{\ell j}(p_i^t). \tag{2}$$

We then run the model on an uninformative ICL prompt $\tilde{p}_i^t \in \tilde{P}_t$ where each $x$ is matched with a *random* output $\tilde{p}_i^t = [(x_i, \tilde{y}_i)]$. Now, the model is less likely to generate the correct output $y_q$ as it cannot infer the relationship from incorrect ICL examples (notwithstanding the observation from Min et al. (2022) that some tasks can be guessed from incorrect labels). While running the model on $\tilde{p}_i^t$, we replace an attention head activation $a_{\ell j}$ with mean task-conditioned activation $\bar{a}_{\ell j}^t$ (Eq. 2) and measure its causal indirect effect (CIE) towards recovering the correct answer $y_q$ as

$$\text{CIE}(a_{\ell j} \mid \tilde{p}_i^t) = f(\tilde{p}_i^t \mid a_{\ell j} := \bar{a}_{\ell j}^t)[y_{iq}] - f(\tilde{p}_i^t)[y_{iq}]. \tag{3}$$

The intuition here is to measure the degree to which using the "correct" mean attention head output $\bar{a}_{\ell j}^t$—computed over the *uncorrupted* prompts for task $t$—increases the mass assigned to the target response $y_{iq}$, relative to the likelihood of this token under the corrupted prompt $\tilde{p}_i^t$. A larger value implies that the corresponding head is more influential in promoting the correct response.

Then each attention head's average indirect effect (AIE) is calculated by averaging this difference across all tasks $t \in \mathcal{T}$ and (corrupted) prompts:

$$\text{AIE}(a_{\ell j}) = \frac{1}{|\mathcal{T}|} \sum_{t \in \mathcal{T}} \frac{1}{|\tilde{P}_t|} \sum_{\tilde{p}_i^t \in \tilde{P}_t} \text{CIE}(a_{\ell j} \mid \tilde{p}_i^t) \tag{4}$$

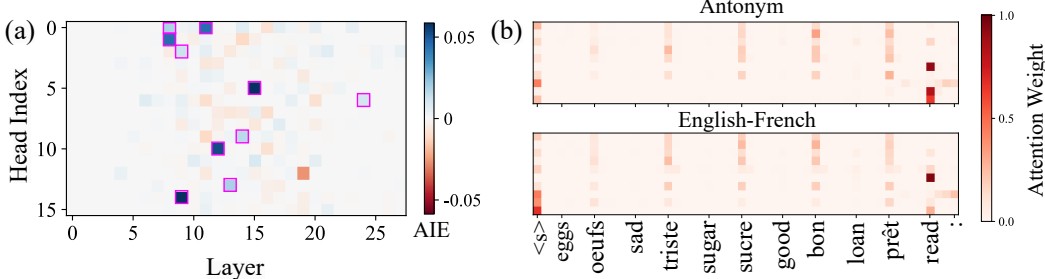

Figure 3: (a) Average indirect effect across all tasks $\mathcal{T}$ for each attention head in GPT-J, and (b) the top 10 heads' weights on individual tokens for one example prompt $p_i^t$. The most strongly implicated heads appear in middle layers. Attention weights are strongest on the output tokens of each exemplar.

To identify the set of attention heads with the strongest causal effects, we repeat this process for each attention head $a_{\ell j}$ in $f$, for all layers $\ell$, and all head indices $j$. We gather the attention heads with highest AIE over all layers as the set $\mathcal{A}$.[1]

Figure 3a shows the AIE per attention head in GPT-J over many tasks (see Appendix G for larger models). The 10 attention heads with highest AIE (which make up $\mathcal{A}$) are highlighted in pink (square outlines) and are clustered primarily in early-middle layers of the network. The average attention pattern of these heads at the final token is shown for two tasks in Figure 3b. These heads primarily attend to token positions corresponding to example outputs; this observation is consistent with the high salience of ICL label tokens observed by Wang et al. (2023a) and while this resembles the same prefix-matching attention pattern as "induction heads" (Elhage et al., 2021; Olsson et al., 2022) not all heads in $\mathcal{A}$ reproduce this pattern on other contexts with repeated tokens (Appendix H).

Due to their high causal influence across many tasks (see Appendix G for breakouts by task), we hypothesize that this small set of heads is responsible for transporting information identifying the demonstrated ICL task. We can represent the contribution of $\mathcal{A}$ as a single vector by taking the sum of their average outputs, over a task, which we call a *function vector* (FV) for task $t$:

$$v_t = \sum_{a_{\ell j} \in \mathcal{A}} \bar{a}_{\ell j}^t \tag{5}$$

We can then test the causal effect of an FV by adding it to hidden states at any layer $\ell$ as the model resolves a prompt and measuring its performance in executing the task (Appendix B).

## 3 EXPERIMENTS

**Models.** We deploy a series of decoder-only autoregressive language models; each is listed and described in Table 1. We use `huggingface` implementations (Wolf et al., 2020) of each model.

**Tasks.** We construct a diverse array of over 40 relatively simple tasks to test whether function vectors can be extracted in diverse settings. To simplify the presentation of our analysis, we focus on a representative sample of 6 tasks:

- **Antonym.** Given an input word, generate the word with opposite meaning.
- **Capitalize.** Given an input word, generate the same word with a capital first letter.
- **Country-Capital.** Given a country name, generate the capital city.
- **English-French.** Given an English word, generate the French translation of the word.
- **Present-Past.** Given a verb in present tense, generate the verb's simple past inflection.
- **Singular-Plural.** Given a singular noun, generate its plural inflection.

All other tasks are described in Appendix E.

---

[1]For GPT-J, we use $|\mathcal{A}| = 10$ attention heads. For larger models, we scale the number of attention heads we use approximately proportionally to the number of attention heads in the model. (We use 20 heads for Llama 2 (7B), 50 for Llama 2 (13B) & GPT-NeoX, and 100 for Llama 2 (70B).)

Table 1: Models used in this study. We focus on decoder-only autoregressive language models that are capable of ICL. For each model, we present the number of parameters, the number of layers $|L|$, and number of attention heads per layer $J = |a_\ell|$.

| Model | Huggingface ID | Citation | Parameters | Training Tokens | $|L|$ | $|a_\ell|$ |
|---|---|---|---|---|---|---|
| GPT-J | EleutherAI/gpt-j-6b | (Wang & Komatsuzaki, 2021) | 6B | 402B | 28 | 16 |
| GPT-NeoX | EleutherAI/gpt-neox-20b | (Black et al., 2022) | 20B | 472B | 44 | 64 |
| Llama 2 | meta-llama/Llama-2-7b-hf | (Touvron et al., 2023) | 7B | 2T | 32 | 32 |
| Llama 2 | meta-llama/Llama-2-13b-hf | (Touvron et al., 2023) | 13B | 2T | 40 | 40 |
| Llama 2 | meta-llama/Llama-2-70b-hf | (Touvron et al., 2023) | 70B | 2T | 80 | 64 |

Table 2: Average accuracy across 6 tasks (macro-averaged across random seeds) for both shuffled-label and zero-shot contexts - adding the FV increases performance of the task compared to the base model in both contexts. For GPT-J we compare to layer averages (Section 2.1) and find that our FV works best. We also report results for both settings on an additional 34 tasks for GPT-J+FV and Llama 2 (70B)+FV. More details on additional tasks in Appendix E.3.

| | Shuffled-Label $[(x_{i1}, \tilde{y}_{i1}), \cdots, (x_{iN}, \tilde{y}_{iN}), x_{iq}]$ | Zero-Shot $[x_{iq}]$ |
|---|---|---|
| GPT-J (baseline on uninformative input) | $39.1 \pm 1.2\%$ | $5.5 \pm 0.8\%$ |
| + $\bar{\mathbf{h}}_\ell^t$ Layer average (Section 2.1) | $79.5 \pm 3.1\%$ | $9.5 \pm 1.8\%$ |
| + $v_t$ FV (Eq. 5) | $\mathbf{90.8} \pm 0.9\%$ | $\mathbf{57.5} \pm 1.7\%$ |
| GPT-NeoX (baseline on uninformative input) | $32.5 \pm 1.3\%$ | $6.7 \pm 0.1\%$ |
| + $v_t$ FV | $\mathbf{90.7} \pm 0.6\%$ | $\mathbf{57.1} \pm 1.5\%$ |
| Llama 2 (70B) (baseline on uninformative input) | $52.3 \pm 2.2\%$ | $8.2 \pm 0.7\%$ |
| + $v_t$ FV | $\mathbf{96.5} \pm 0.5\%$ | $\mathbf{83.8} \pm 0.7\%$ |
| GPT-J + $v_t$ FV on 34 additional tasks | $80.4 \pm 0.6\%$ | $46.1 \pm 3.7\%$ |
| Llama 2 (70B) + $v_t$ FV on 34 additional tasks | $93.0 \pm 0.5\%$ | $74.2 \pm 3.1\%$ |

## 3.1 PORTABILITY OF FUNCTION VECTORS

In this section, we investigate the portability of function vectors—i.e., the degree to which adding an FV to a particular layer at the final token position of the prompt can cause the language model to perform a task in contexts that differ from the ICL contexts from which it was extracted. For simplicity of analysis, we only include test queries for which the LM answers correctly given a 10-shot ICL prompt; all accuracies and standard deviations over 5 random seeds are reported on this filtered subset, and can be thought of as the proportion of model's task performance encoded by FVs. Results when incorrect ICL are included are similar (see Appendix D).

**Evaluating FVs at Layer** $|L|/3$. In Table 2 we report results (averaged across the 6 tasks mentioned above) for adding FVs to shuffled-label ICL prompts and zero-shot contexts across 3 models - GPT-J, GPT-NeoX and Llama 2 (70B), at layers 9, 15, and 26 respectively (approximately $|L|/3$). For GPT-J, we also compare the efficacy of FVs to other approaches for extracting task-inducing vectors including simple state averaging (§2.1).

Our first observation is that the base model is substantially unable to perform the tasks in the uninformative shuffled-label ICL and zero-shot settings; however, adding the FV allows the model to recover task performance significantly in both cases. We also observe the proposed approach for constructing FVs via causal mediation outperforms the layer-averaging $\bar{\mathbf{h}}_\ell^t$ approach in both contexts.

**Zero-Shot Results Across Layers.** Figure 4 shows results across layers for the zero-shot case. The sharp reduction of causal effects in late layers suggests that FVs do not simply act linearly, but that they trigger late-layer nonlinear computations. This pattern of causality is seen across a variety of tasks, autoregressive model architectures, and model sizes. Even in cases where performance is low, as in English-French with GPT-NeoX and Llama 2 (70B), adding the function vector in middle layers still results in large *relative* improvements to accuracy over the zero-shot baseline. Results are also consistent across model sizes: see Appendix J for results with all sizes of Llama 2.

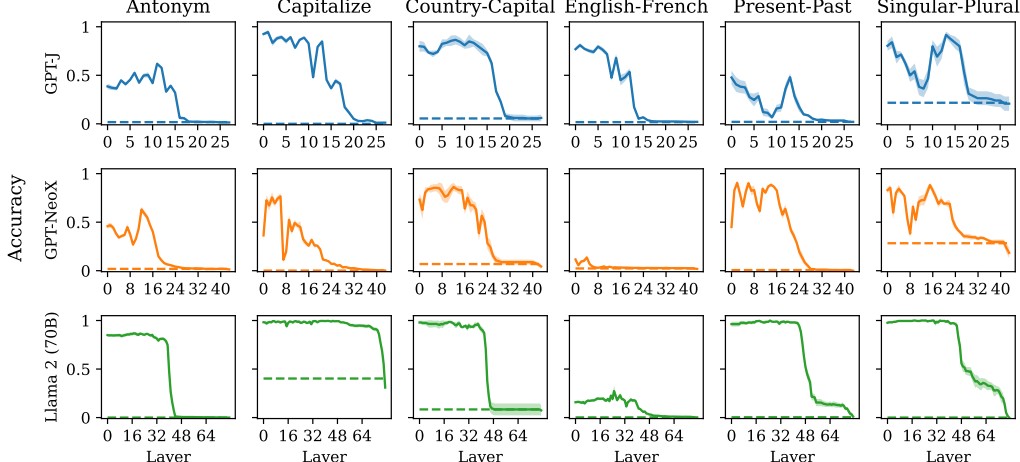

Figure 4: Task accuracy across tasks and models, applying FVs in zero-shot settings. We show accuracies before adding the function vector (dotted lines) and after adding the FV to a specific layer (solid lines). Adding the FV to early-middle layers pushes models to perform the target task without any exemplars, as demonstrated by accuracy increases over the zero-shot without FVs.

Table 3: Natural text portability of the Antonym FV. We provide a natural template and substitute in a query word for 'x'. Then, we measure accuracy based on whether the correct antonym is produced in this natural text setting within 5 generated tokens.

| Prompt | GPT-J | +Antonym FV |
|---|---|---|
| The word "x", means | $1.5 \pm 1.1\%$ | $55.2 \pm 3.8\%$ |
| When I think of the word "x", it usually means | $0.3 \pm 0.2\%$ | $67.7 \pm 3.0\%$ |
| When I think of x, I usually | $0.0 \pm 0.0\%$ | $61.1 \pm 2.4\%$ |
| While reading a book, I came across the word "x". I looked it up in a dictionary and it turns out that it means | $2.7 \pm 1.9\%$ | $46.0 \pm 4.6\%$ |
| The word x can be understood as a synonym for | $2.4 \pm 1.7\%$ | $52.7 \pm 11.0\%$ |

**FVs are Robust to Input Forms.**    To check whether the FV is dependent on the ICL template that it is extracted from, we also test the FV on 20 additional ICL templates (Appendix C) and in natural text settings, adding the FV at layer $\ell = 9$ for GPT-J (approximately $|L|/3$).

We create 20 different ICL templates that vary the form of the ICL prompt across prefixes and delimiters of input-output pairs. We evaluate FVs on GPT-J for these 20 templates in both shuffled-label and zero-shot settings. Across our 6 tasks, adding the FV executes the task with an average accuracy of $76.2 \pm 13.8\%$ with shuffled labels and $40.0 \pm 16.7\%$ in the zero-shot setting, while GPT-J only scores $32.3 \pm 12.8\%$ and $6.2 \pm 4.3\%$ on the same settings, respectively. Despite higher variance, this performance is similar to performance in the same settings with the original template.

We also evaluate FVs on natural text completions. Given a natural text template, we insert a test query word and have the model generate $n$ tokens. We add the FV to the final token of the original prompt, and for all subsequent token predictions to guide its generation. We use a simple regex match to compute whether the generation includes the correct target for the inserted query word.

Table 3 shows natural text portability results for the antonym FV for GPT-J, generating 5 new tokens. In each of the templates, the antonym is in the FV completion significantly more than the original completion. In fact, we find that the efficacy of the antonym FV in eliciting the correct response in these natural text templates performs on par with the results previously reported for the zero-shot setting. This is true for all 6 tasks (Appendix F), suggesting that the task representation transported during ICL is similar to one that is used during autoregressive prediction in natural text settings.

We include a few qualitative results for the English-French and Country-Capital tasks (Table 4). We see that the English-French FV will sometimes translate the whole sentence after giving the proper completion to the original one-word translation task, indicating that it has captured more than the original task it was shown. Additional natural text portability results are included in Appendix F.

Table 4: Qualitative examples of natural text completions for English-French, and Country-Capital

| **English-French** | | |
|---|---|---|
| **Prompt:** | The word "daily" means | The word 'link' can be understood as a synonym for |
| **GPT-J** | every day | 'connection' or 'relation'. The term 'link' is used in... |
| **GPT-J+English-French FV** | tous les jours | 'lien', et le mot 'lien' peut être compris comme un synonyme... |
| **Country-Capital** | | |
| **Prompt:** | When you think of Netherlands, | |
| **GPT-J** | you probably think of tulips, windmills, and cheese. But the Netherlands is also home to... | |
| **GPT-J+Country-Capital FV** | you think of Amsterdam. But there are many other cities in the Netherlands. Here are some... | |

Table 5: A direct decoding of the function vector for each task.

| **Task $t$** | **Tokens in the distribution $D(v_t)$ in order of decreasing probability** |
|---|---|
| Antonym | ' lesser', ' counterpart', 'wrong', ' negate', ' destroy' |
| Capitalize | ' Vanilla', ' Copy', ' Adapter', ' Actor', ' Container' |
| Country-Capital | ' Moscow', ' Bangkok', ' Paris', ' London', ' Madrid' |
| English-French | ' âĶĬ', ' masc', ' ç¥l', ' embr', ' è' |
| Present-Past | 'received', 'changed', 'killed', 'answered', ' Changed' |
| Singular-Plural | 'cards', 'stocks', ' helmets', ' items', ' phones' |

## 3.2 THE DECODED VOCABULARY OF FUNCTION VECTORS

Several studies have gleaned insights about the states and parameters of transformers by viewing them in terms of their decoded vocabulary tokens (Nostalgebraist, 2020; Geva et al., 2021; 2022; Dar et al., 2023; Belrose et al., 2023). Therefore we ask: can we understand an FV by decoding $v_t$ directly to a token probability distribution? Results are shown in Table 5, which lists the top five tokens in the decoded distribution $D(v_t)$ for each task (additional tasks in Appendix I).

A clear pattern emerges: for most tasks, the decoded tokens lie within the task's output space. The Singular-Plural function vector decodes to a distribution of plural nouns, and Present-Past decodes to past-tense verbs. However, that is not the case for all tasks: English-French decodes to nonsense tokens, and the Antonym task decodes to words that evoke the abstract idea of reversal.

Given these meaningful decodings, we then ask whether the token vocabulary is sufficient to recreate a working function vector. That is, we begin with the token distribution $Q_t = D(v_t)$, and determine whether a function vector can be reconstructed if we know the top words in $Q_t$. Denote by $Q_{tk}$ the distribution that resamples $Q_t$ while restricting to only the top $k$ words. We perform an optimization to reconstruct a $\hat{v}_{tk}$ that matches the distribution $Q_{tk}$ when decoded (where CE is cross-entropy loss):

$$\hat{v}_{tk} = \arg\min_v \text{CE}(Q_{tk}, D(v)) \qquad (6)$$

In Table 6, the performance of $\hat{v}_{tk}$ is evaluated when used as a function vector. We find that, while it is possible to partially recreate the functionality of an FV, good performance typically requires more

Table 6: Performance of FV $v_t$ is compared to the reconstruction $\hat{v}_{t\,100}$ that matches the top 100 tokens, and $\hat{v}_{t\,\text{all}}$ that matches all 50k tokens in $D(v_t)$. The KL divergence between the $D(\hat{v}_{tk})$ and $Q_{tk}$ are shown for each reconstruction as $\text{KL}_k$. Lowest performers for each task in red.

| | Accuracy on zero-shot prompts | | | | |
|---|---|---|---|---|---|
| **Task $t$** | $v_t$ | $\hat{v}_{t\,100}$ | $\text{KL}_{100}$ | $\hat{v}_{t\,\text{all}}$ | $\text{KL}_{\text{all}}$ |
| Antonym | $48.2 \pm 2.0\%$ | $4.8 \pm 2.0\%$ | 0.0033 | $39.6 \pm 2.6\%$ | 0.0137 |
| Capitalize | $70.5 \pm 2.4\%$ | $5.7 \pm 2.2\%$ | 0.0001 | $51.5 \pm 11.6\%$ | 0.0053 |
| Country-Capital | $83.2 \pm 2.7\%$ | $58.1 \pm 18.5\%$ | 0.0002 | $29.0 \pm 15.1\%$ | 0.0019 |
| English-French | $44.7 \pm 1.2\%$ | $4.8 \pm 1.7\%$ | 0.0 | $42.0 \pm 5.6\%$ | 0.0056 |
| Present-Past | $19.7 \pm 5.9\%$ | $4.4 \pm 1.4\%$ | 0.0052 | $6.8 \pm 2.6\%$ | 0.0139 |
| Singular-Plural | $47.0 \pm 3.4\%$ | $23.3 \pm 6.1\%$ | 0.0 | $27.4 \pm 4.7\%$ | 0.0145 |

than 100 vocabulary tokens. In other words, knowledge of the top decoded tokens of $D(v_t)$ is usually not enough on its own to construct a working function vector. That suggests that the FV contains some needed information beyond that expressed by its top decoded tokens.

## 3.3 VECTOR ALGEBRA ON FUNCTION VECTORS

(a) Input: "Italy, Russia, China, Japan, France"

(b)

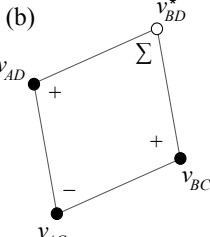

| FV | Task | Expected Output |
|---|---|---|
| $v_{AC}$ | First-Copy | Italy |
| $v_{AD}$ | First-Capital | Rome |
| $v_{BC}$ | Last-Copy | France |
| $v^*_{BD}$ | Last-Capital | Paris |

Figure 5: (a) A set of three list-oriented tasks that can be composed to a fourth task using FV vector algebra. (b) The parallelogram arrangement of the fourth vector $v^*_{BD}$ when it is composed out of the other three FVs.

Although Table 6 suggests that function vectors cannot be understood as simple semantic vector offsets on word embeddings, we can ask whether function vectors obey semantic vector algebra over the more abstract space of *functional* behavior by testing the composition of simple functions into more complex ones. We begin with three conceptually decomposable ICL tasks: the list-oriented tasks First-Copy, First-Capital, and Last-Copy, as illustrated in Figure 5a. Using ICL, we collect FVs for all three tasks and denote them $v_{AC}$, $v_{BC}$, and $v_{AD}$.

Then we form a simple algebraic sum to create a new vector that we will denote $v^*_{BD}$.

$$v^*_{BD} = v_{AD} + v_{BC} - v_{AC} \tag{7}$$
$$\text{Last-Capital}^* = \text{Last-Copy} + \text{First-Capital} - \text{First-Copy} \tag{8}$$

In principle we could expect $v^*_{BD}$ to serve as a new function vector for a new composed task (Last-Capital). We perform several similar task compositions on a variety of tasks. In each case, we combine a task with First-Copy and Last-Copy to produce a composed Last-$*$ vector; then, we test the accuracy of $v^*_{BD}$ as a function vector. We compare to the accuracy of the FV extracted from ICL, as well as accuracy of the same model performing the task using ICL. Results for GPT-J are reported in Table 7; see Appendix K for results for Llama 2 (13 and 70 billion parameter models). We find that some FVs can be composed, with algebraic compositions outperforming FVs and even ICL on some tasks. Other tasks, including some for which ICL and FVs perform well, resist vector composition.

The ability to compose the tasks that we have demonstrated may hinge on the fact that "word-selection" from context and "word-transformation" are different components of language tasks that could involve FVs triggering complementary underlying mechanisms (e.g., one for locating and extracting input and another for transforming it). We therefore believe that FV composition may be a useful tool for further understanding the mechanisms of LMs.

Table 7: The accuracy of ICL, calculated FV $v_{BD}$ zero-shot interventions, and vector-composed $v^*_{BD}$ zero-shot interventions when performing several list-oriented tasks. Unlike our previous evaluations, here we measure performance on *all* available samples of the task, without restriction to the subset where the LM predicts correct output. In a few cases, composed function vector intervention $v^*_{BD}$ can perform a task better than ICL.

| Task | ICL (ten-shot) | $v_{BD}$ (FV on zero-shot) | $v^*_{BD}$ (sum on zero-shot) |
|---|---|---|---|
| Last-Antonym | $0.25 \pm 0.02$ | $0.02 \pm 0.01$ | $0.07 \pm 0.02$ |
| Last-Capitalize | $0.91 \pm 0.02$ | $0.64 \pm 0.03$ | $0.76 \pm 0.04$ |
| Last-Country-Capital | $0.32 \pm 0.02$ | $0.15 \pm 0.03$ | $\mathbf{0.60} \pm 0.02$ |
| Last-English-French | $0.45 \pm 0.04$ | $0.16 \pm 0.02$ | $0.06 \pm 0.02$ |
| Last-Present-Past | $0.89 \pm 0.02$ | $0.18 \pm 0.02$ | $0.29 \pm 0.03$ |
| Last-Singular-Plural | $0.90 \pm 0.01$ | $0.28 \pm 0.01$ | $0.29 \pm 0.02$ |
| Last-Capitalize-First-Letter | $0.75 \pm 0.01$ | $0.76 \pm 0.02$ | $\mathbf{0.95} \pm 0.00$ |
| Last-Product-Company | $0.35 \pm 0.03$ | $0.30 \pm 0.02$ | $\mathbf{0.41} \pm 0.03$ |

## 4 RELATED WORK

A cousin to function vectors has been independently observed in concurrent work by Hendel et al. (2023); they study causal effects of $\mathbf{h}_\ell^t$ (similar to Section 2.1) on a different set of models and tasks.

**Task Representations.** Our work shows that it is possible to extract FVs with strong causal effects from LLMs; this is an advance over previous examinations that have *added* task representations to LLMs, e.g. Lampinen & McClelland (2020); Shao et al. (2023); Mu et al. (2023); Panigrahi et al. (2023); Ilharco et al. (2023), who devised ways to create compositional task encodings for LLMs using metamappings, codebooks, soft-prompts or sets of model parameter perturbations that Ilharco et al. calls *task vectors*. Unlike these previous works that create function representations, we find that compact FVs *already exist* within LLMs and show how to extract them. Likewise Lake & Baroni (2018); Hill et al. (2018) show RNN hidden states cluster on similar tasks. Our work differs because FVs are causal, not just correlative, so they can be explicitly extracted and inserted.

**In-Context Learning.** Since its observation in LLMs by Brown et al. (2020), ICL has been studied intensively from many perspectives. The role of ICL prompt forms has been studied by Reynolds & McDonell (2021); Min et al. (2022); Yoo et al. (2022). Models of inference-time metalearning that could explain ICL have been proposed by Akyürek et al. (2022); Dai et al. (2023); Von Oswald et al. (2023); Li et al. (2023b); Garg et al. (2022). Analyses of ICL as Bayesian task inference have been performed by Xie et al. (2021); Wang et al. (2023c); Wies et al. (2023); Hahn & Goyal (2023); Zhang et al. (2023); Han et al. (2023). And ICL robustness under scaling has been studied by Wei et al. (2023); Wang et al. (2023b); Pan et al. (2023). Our work differs from those studies of the externally observable behavior of ICL by instead focusing on mechanisms *within* transformers.

**Mechanisms of task performance in LMs.** Our work is related to Merullo et al. (2023); Halawi et al. (2023) which analyze components during execution of ICL tasks and identify causes of false statements. Also related are several methods that modify activations at inference time to steer LM behavior (Li et al., 2023a; Hernandez et al., 2023a; Subramani et al., 2022; Turner et al., 2023; Rimsky et al., 2023; Liu et al., 2023; Zou et al., 2023). Our work is consistent with Wang et al. (2023a) which observes salience of label tokens during ICL, Wang et al. (2022b) which observes individual neurons that correlate with specific task performance, and Variengien & Winsor (2023) which task requests are processed in middle layers. We measure causal mediators across a distribution of different tasks to find a generic function-invocation mechanism that *identifies and distinguishes* between tasks.

**Mechanistic Interpretability.** We also build upon the analyses of Elhage et al. (2021) and Olsson et al. (2022), who observed prevalent in-context copying behavior related to jumps in performance during training. We isolate FVs using causal mediation analysis methods developed in Pearl (2001); Vig et al. (2020); Meng et al. (2022); Wang et al. (2022a); Geva et al. (2023). Our examination of FVs in vocabulary uses the logit lens of Nostalgebraist (2020); Geva et al. (2021); Dar et al. (2023).

**Analyzing the Attention Mechanism.** Our work is related to previous attention-weight analyses (Voita et al., 2018; Clark et al., 2019; Voita et al., 2019; Kovaleva et al., 2019; Reif et al., 2019; Lin et al., 2019; Htut et al., 2019; Kobayashi et al., 2020), that have found attention weights that align with linguistic structures. Our work is motivated by the observation that attention weights alone do not fully explain model outputs (Jain & Wallace, 2019; Wiegreffe & Pinter, 2019; Bibal et al., 2022). The focus of our paper is to extend our understanding of attention by investigating the *content* of the information transported by the attention heads in ICL to open a new window into the human-interpretable role that attention plays in language processing.

## 5 DISCUSSION

Function vectors are a surprising finding. The metalearning capabilities of LLMs that have been studied since Brown et al. (2020) seem complex enough be inscrutable. Yet in this paper we have found a simple mechanism in a range of transformer LLMs that is common across tasks and robust to shifts in context: *function vectors* (FVs) that represent the task within a hidden state. FVs can be explicitly extracted from a small fixed set of attention heads that can be easily identified, and these FVs represent a range of tasks just as simply as word vectors (Mikolov et al., 2013)—yet our findings also reveal FVs must be a distinct phenomenon (Appendix A). Although FVs are not yet a complete accounting of how ICL works, they do provide new clarity on one level of mediation within ICL, and they open up a new path for future research to fully characterize function execution within LLMs.

ETHICS

While our work clarifying the mechanisms of function representation and execution within large models is intended to help make large language models more transparent and easier to audit, understand, and control, we caution that such transparency may also enable bad actors to abuse large neural language systems, for example by injecting or amplifying functions that cause undesirable behavior.

ACKNOWLEDGMENTS

Special thanks to Evan Hernandez whose valuable advice and mentorship made this research possible. We are grateful for the generous support of Open Philanthropy (ET, AS, AM, DB) as well as National Science Foundation (NSF) grant 1901117 (ET, ML, BW). ML is supported by an NSF Graduate Research Fellowship, and AM is recipient of the Zuckerman Postdoctoral Fellowship. We thank the Center for AI Safety (CAIS) for making computing resources available for this research.

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

## A  DISCUSSION: FUNCTION VECTORS VS SEMANTIC VECTOR ARITHMETIC

In this appendix we discuss the experimental support for our characterization of function vectors in more detail, in particular the assertion that function vectors are acting in a way that is distinct from semantic vector arithmetic on word embeddings.

The use of vector addition to induce a mapping is familiar within semantic embedding spaces: the vector algebra of *semantic vector offsets* has been observed in many settings; for example, word embedding vector arithmetic was clearly described by Mikolov et al. (2013), and has been observed in other neural word representations including transformers (recently, Merullo et al., 2023). In recent examinations of internal transformer states, Geva et al. (2022) and Dar et al. (2023) have suggested that many internal transformer calculations can be understood in terms of such word vector arithmetic. Therefore one of our main underlying research questions is whether our function vectors should be described as triggers for a nontrivial function, or whether, more simply (as would be suggested by Dar et al.), they could be thought of as just ordinary semantic vector offsets that induce a trivial mapping between related words by adding an offset to a rich word embedding.

Our investigation of the possibility of a distinct and localized representation for tasks is similar to the studies of task encodings in RNNs by Lake & Baroni (2018) and Hill et al. (2018), as well as the studies of linguistic attribute encodings by Clark et al. (2019), but our focus on causal effects rather than correlations allows us to explicitly extract and test the computational roles of vector representations of functions, which leads to several new lines of evidence.

The main paper contains three pieces of experimental evidence that support the conclusion that function vectors are different from semantic vector offsets of word embeddings, and that they trigger nontrivial functions:

1. Function vectors can implement complex mappings, including cyclic mappings such as *antonyms* that cannot be semantic vector offsets.

2. Function vector causal effects cannot be recovered from the target output vocabulary alone; they carry some other information.

3. Function vector activity is mediated by mid-layer nonlinearities (i.e., they trigger nonlinear computations), since they have near-zero causal effect at late layers.

We discuss each of these lines of evidence in more detail here.

**Cyclic Tasks Cannot be Semantic Vector Offsets.**  The first task analyzed in the paper is the *antonym* task. Because the antonym task is cyclic, it is a simple counterexample to the possibility that function vectors are just semantic vector offsets of language model word embeddings.

Suppose there were an 'antonym' vector offset $v_a$ such that adding it to a word embedding $w$ produces $w'$, the embedding of the antonym of $w$ (i.e. $w + v_a = \mathrm{antonym}(w) = w'$). An example of this might be vec(big) + $v_a$ = vec(small). But then: if $w$ and $w'$ are antonyms of each other, the relationship holds both ways. That means that the antonym offset vector $v_a$ would have to satisfy both $w + v_a = w'$ and $w' + v_a = w$, which can only happen if $v_a = 0$, implying $w = w'$ and creating a contradiction to the assumption that $w$ and $w'$ are antonyms, distinct from each other. Thus there could be no constant vector offset that would properly model the antonym task. The same argument excludes semantic vector offsets for any cyclic function. We evaluate additional cyclic tasks in Appendix L.

Since we are able to find a constant antonym function vector $v_t$ that, when added to the transformer, does cause cyclic behavior, we conclude that the action of $v_t$ is a new phenomenon. Function vectors act in a way different from simple semantic vector offsets.

**Function Vectors Contain Information Beyond Output Vocabulary.**  Not every function is cyclic, but the vector offset hypothesis can be tested by examining word embeddings. Following the reasoning of Geva et al. (2022); Dar et al. (2023), one way to potentially implement a semantic vector offset is to promote a certain subset of tokens that correspond to a particular semantic concept (i.e. capital cities, past-tense verbs, etc.). Function vectors do show some evidence of acting in this way: when decoding function vectors directly to the model's vocabulary space we often see that the tokens

with the highest probabilities are words that are part of its task's output vocabulary (see Table 5 and Table 19).

To determine whether it is the vocabulary itself that contributes to the function vector's performance, in Section 3.2 we construct another vector (via optimization) that decodes to the same vocabulary distribution and measure its performance when used as a "function vector". In that experiment we create reconstructions $\hat{v}_{t100}$ and $\hat{v}_{t\text{all}}$ that encode the same decoder vocabulary as $v_t$ (the near-zero KL divergences show that the reconstructions of the decoder vocabulary are near-perfect). Yet the performance of the reconstructions when used as FV is poor when the top 100 words are matched, and still lower than $v_t$ even when the distribution over the full vocabulary is reconstructed.

That experiment reveals that while function vectors do often encode words contained in the output space of the task they are extracted from, simply adding a vector that boosts those same words by the same amounts is not enough to recover its full performance—though if enough words are included, in some cases a fraction of performance is recovered. Our measurements suggest that while *part* of the role of a function vector may act similarly to a semantic vector offset, the ability for function vectors to produce nontrivial task behavior arises from other essential information in the vector beyond just a simple word embedding vocabulary-based offset. See Appendix N for a related experiment.

**Function Vectors' Causal Effects are Near-Zero at Late Layers.**   Across the set of tasks and models we evaluate, there is a common pattern of causal effects that arises when adding a function vector to different layers. The highest causal effects are achieved when adding the function vector at early and middle layers of the network, with a sharp drop in performance to near-zero at the later layers of the network (Figure 4, Figure 26, Figure 14, Figure 16). Interestingly, FV causal effects are strongest in largest models, yet the cliff to near-zero causal effects is also sharpest for the largest models (Figure 16).

If the action were linear and created by a word embedding vector offset, then the residual stream would transport the action equally well at any layer including the last layers. Thus the pattern of near-zero causal effects at later layers suggests that the action of the function vector is not acting directly on the word embedding, but rather that it is mediated by some *nonlinear* computations in the middle layers of the network that are essential to the performance of the task. This mediation is evidence that the function vector activates mid-layer components that execute the task, rather than fully executing the task itself.

This pattern is in contrast to the vector arithmetic described in Merullo et al. (2023), that are most effective at later layers of the network and have little to no causal effect at early layers of the network; those offsets more closely resemble semantic vector offsets of word embeddings.

**Function Vectors Therefore Represent Functions**   In summary, the three lines of evidence lead us conclude that the vectors $v_t$ should not be seen as simple word embeddings, nor trivial offsets or differences of embeddings, nor simple averages of word embeddings over vocabularies of words to be boosted. These characteristics distinguish function vectors from from many linguistic concepts that can be viewed as probability distributions over words. Rather, the evidence suggests that the vectors we have identified act in a way that is distinct from literal token embedding offsets. Instead they directly represent and trigger nonlinear execution of abstract functions.

This finding is surprising since the transformer is trained on a word-prediction task; it would be expected that they should learn representations that can be expressed in terms of adjustments to word probabilities as observed by Geva et al. (2022); Dar et al. (2023). Our evidence indicates a different kind of representation: we find that transformers learn a compact, concrete, and causal vector representation of higher-level functional concepts that cannot be reduced to a probability vector over a set of words.

Thus we come to the conclusion that the vectors $v_t$ are references to functions, and we call them *function vectors*.

## B  ATTENTION OUTPUTS ARE INDEPENDENT AND ADDITIVE

In this section we define our formulation of attention notation $a_{\ell j}$ in detail, relating our notation to the original formulation of Vaswani et al. (2017) via the framework of Elhage et al. (2021).

### B.1  EXPRESSING ATTENTION $a_{\ell j}$ IN THE RESIDUAL STREAM SPACE

The transformer architecture as introduced by Vaswani et al. (2017) has a multihead attention mechanism. They describe it in terms of a concatenation procedure for performing the attention function for several "heads" $\text{head}_j \in \mathbb{R}^{d_v}$ ($j \leq J$) at each layer, all in parallel. Equation 9 reproduces the relevant equation from page 5 of Vaswani et al.:

$$\text{MultiHead}(Q, K, V) = \text{Concat}(\text{head}_1, ..., \text{head}_h)W^O \tag{9}$$

$$\text{where } \text{head}_j = \text{Attention}(QW_j^Q, KW_j^K, VW_j^V) \tag{10}$$

Note that a transformer repeats this process with different weights and data at each layer; we add the layer subscript $\ell$ to disambiguate their notation. In the Vaswani et al. formulation, each head at a layer $\ell$ resides in a low dimensional space $\mathbb{R}^{d_v}$ with dimension $d_v < d$ that differs from the main hidden state *residual stream* of the transformer, which we write as $\mathbf{h}_\ell \in \mathbb{R}^d$. All the heads at the layer are concatenated and then transformed through transformation $W_\ell^O$ to produce the full $\text{MultiHead}_\ell$ attention output in $\mathbb{R}^d$.

Elhage et al. (2021) observes that the Vaswani et al. formulation is equivalent to dividing the matrix $W_\ell^O$ into block form $[W_{\ell 1}^O \, W_{\ell 2}^O \ldots \, W_{\ell J}^O]$, and then projecting each $\text{head}_{\ell j}$ into the residual stream space directly. In our formulation of $a_{\ell j}$ in Section 2.2, we adopt this view. The attention head output $a_{\ell j}$ can be defined in terms of the notation of Vaswani et al. and Elhage et al. as follows:

$$a_{\ell j} = \text{head}_{\ell j} W_{\ell j}^O \in \mathbb{R}^d \tag{11}$$

In this way the total attention contribution at layer $\ell$ is the sum of the attention output of each individual head, and these all reside directly in the $\mathbb{R}^d$ space of the residual stream:

$$\text{MultiHead}_\ell(Q_\ell, K_\ell, V_\ell) = \sum_{j \leq J} a_{\ell j} \in \mathbb{R}^d \tag{12}$$

While the left and right-hand sides of equation 12 are computationally equivalent definitions of attention, the "independently additive" form of attention on the right-hand side allows us to see the contributions of individual attention heads more clearly.

### B.2  ADDING FUNCTION VECTORS TO A LAYER

Using this formulation of attention $a_{\ell j}$, we return to our notation as defined in section 2.2 to understand what we mean when we say we add a function vector to a layer $\ell$.

We focus on the hidden state residual stream at the final token of a given prompt. Recall that the ordinary operation of a transformer defines the hidden state

$$\mathbf{h}_\ell = \mathbf{h}_{\ell-1} + m_\ell + \sum_{j \leq J} a_{\ell j} \tag{13}$$

This recursive residual structure forms a telescoping sum that creates a common vector space for nearby layers; for example Zhao et al. (2021) has observed that adjacent layers of a transformer can be swapped with little change. Thus it is meaningful to collect, average, and sum attention head outputs $a_{\ell j} \in \mathbb{R}^d$ among nearby layers, and that observation inspires our definition of a function vector $v_t$ as average attention head values for a selection of relevant heads $\bar{a}_{\ell j}$ (equation 5). See Appendix M for an analysis of an alternative formulation that does not swap layers.

Since the function vector also resides in the residual stream space of $\mathbf{h}_\ell \in \mathbb{R}^d$, when we add a function vector $v_t$ to the hidden state of layer $\ell$, we can therefore add it to the hidden state just as attention is added; we write the updated hidden state $\mathbf{h}_\ell'$ as

$$\mathbf{h}_\ell' = \mathbf{h}_{\ell-1} + m_\ell + \sum_{j \leq J} a_{\ell j} + v_t \tag{14}$$

This could also be written as simply $\mathbf{h}_\ell' = \mathbf{h}_\ell + v_t$.

## C  EXPERIMENTAL DETAILS

In this section, we provide details of the function vector extraction process (section 2.3), and the evaluation of function vectors (section 3).

**Function Vector Extraction.**  We compute a function vector as the sum over the average output of several attention heads, where the average is conditioned on prompts taken from a particular task. We write this as $v_t = \sum_{a_{\ell j} \in \mathcal{A}} \bar{a}^t_{\ell j}$. How many attention heads should we use? To extract a function vector (FV), we first compute the task-conditioned mean activation of each head $\bar{a}^t_{\ell j}$, using $|P_t| = 100$ clean (uncorrupted) 10-shot prompts. We use this to identify a set of causal attention heads $\mathcal{A}$, which are ranked based on the average indirect effect (AIE) of each head. For GPT-J we found that the increase in performance for many tasks begins to plateau when using $|\mathcal{A}| = 10$ attention heads, though for some tasks using more heads increases performance even more (Figure 6).

The AIE is computed over a subset of all abstractive tasks (Appendix E), using $|\tilde{P}_t| = 25$ corrupted 10-shot prompts per task. Because we are interested in tasks the model can successfully do via ICL, a task is only included if its 10-shot ICL performance is better than the baseline (majority-label) performance. For GPT-J, there are $|\mathcal{T}| = 18$ tasks satisfying this criteria which we use to compute the AIE (Figure 8).

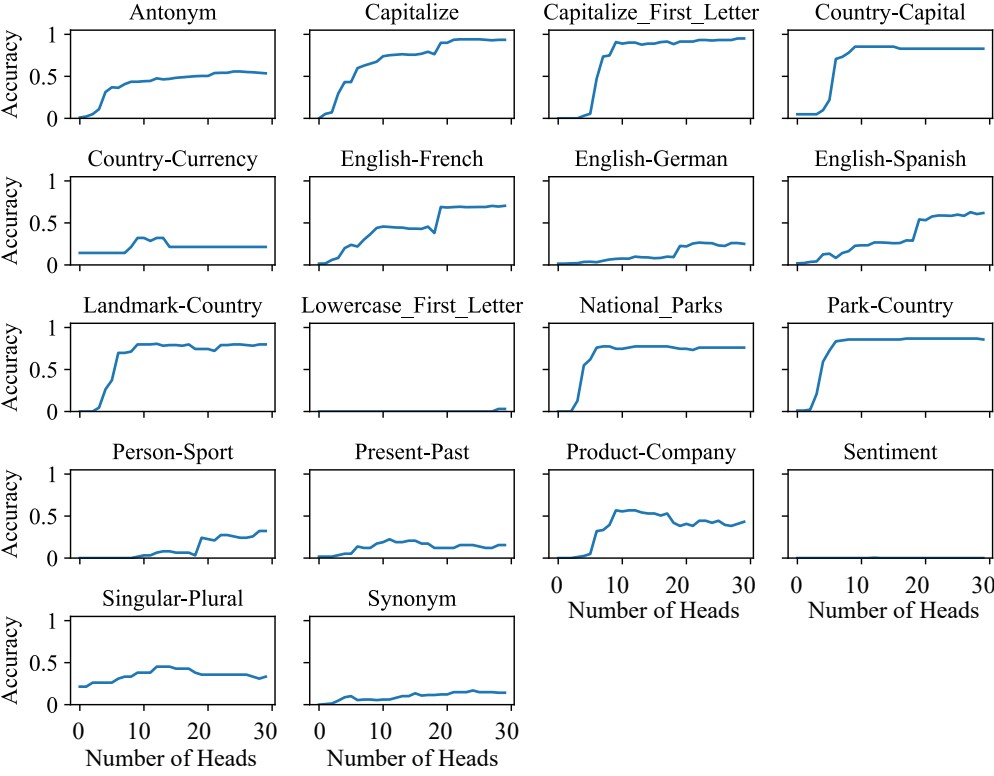

Figure 6: Zero-shot accuracy across 18 different tasks for adding a function vector to GPT-J. We vary the number of heads in $\mathcal{A}$ that are used to create the function vector and find that the change in performance begins to plateau around $|\mathcal{A}| = 10$ attention heads for a majority of the tasks. For this reason, we use $|\mathcal{A}| = 10$ for GPT-J.

**Evaluating Function Vectors.**  To evaluate a function vector's (FV) causal effect, we add the FV to the output of a particular layer in the network ($\mathbf{h}_\ell$) at the last token of a prompt $p$ with query $x_q$, and then measure whether the predicted word matches the expected answer $y_q$. This can be expressed simply as $f(p \mid \mathbf{h}_\ell := \mathbf{h}_\ell + v_t)$ (see section B for a more detailed explanation). We report this top-1

accuracy score over the test set under this intervention. If $y_q$ is tokenized as multiple tokens, we use the first token of $y_q$ as the target token.

For all results in Section 3 and in the following appendix sections (unless stated otherwise - e.g. Figure 4), we add the FV to the hidden state at layer $\ell \approx |L|/3$, which we found works well in practice. This corresponds to layer 9 for GPT-J, layer 15 for GPT-NeoX, layer 11 for Llama 2 (7B), layer 14 for Llama 2 (13B) and layer 26 for Llama 2 (70B).

**Prompt Templates.** The default template we use to construct ICL prompts is: Q:$\{x_{ik}\}$\nA:$\{y_{ik}\}$\n\n, where $x_{ik}$ and $y_{ik}$ (or $\tilde{y}_{ik}$ for corrupted prompts) are substituted for the corresponding element in brackets, and each example is concatenated together. An example of a full prompt template is:

$$Q:\{x_{i1}\}\backslash nA:\{y_{i1}\}\backslash n\backslash n \ \ldots \ Q:\{x_{iN}\}\backslash nA:\{y_{iN}\}\backslash n\backslash nQ:\{x_{iq}\}\backslash nA: \tag{15}$$

To evaluate a function vector we use a few different prompt contexts. The shuffled-label prompts are corrupted 10-shot prompts with the same form as (15), while zero-shot prompts only contain a query $x_q$, without prepended examples (e.g. Q:$\{x_{iq}\}$\nA:).

In section 3.1 we use FVs extracted from prompts made with the template shown in (15), and test them across a variety of other templates (Table 8).

Table 8: We test a variety of ICL prompt templates in §3.1, which are shown below. The function vectors (FVs) we collect are constructed from a default template of the form Q:$\{x_{ik}\}$\nA:$\{y_{ik}\}$\n\n, and tested on prompts created with the new prompt form.

| Template Forms | |
| --- | --- |
| question:$\{x_{ik}\}$\n answer:$\{y_{ik}\}$\n\n, | question:$\{x_{ik}\}$\n answer:$\{y_{ik}\}$\|, |
| A:$\{x_{ik}\}$\n B:$\{y_{ik}\}$\n\n, | Question:$\{x_{ik}\}$\n\n Answer:$\{y_{ik}\}$\n\n, |
| Input:$\{x_{ik}\}$ Output:$\{y_{ik}\}$\|, | $\{x_{ik}\}$\n $\rightarrow\{y_{ik}\}$\n\n, |
| $\{x_{ik}\}$\n :$\{y_{ik}\}$\n , | input:$\{x_{ik}\}$ output:$\{y_{ik}\}$, |
| question:$\{x_{ik}\}$ answer:$\{y_{ik}\}$, | x:$\{x_{ik}\}$\n y:$\{y_{ik}\}$\|, |
| input:$\{x_{ik}\}$\|output:$\{y_{ik}\}$\n, | $\{x_{ik}\}$ $\rightarrow\{y_{ik}\}$\n\n, |
| input:$\{x_{ik}\}$ output:$\{y_{ik}\}$\n, | In:$\{x_{ik}\}$ Out:$\{y_{ik}\}$\|, |
| text:$\{x_{ik}\}$\|label:$\{y_{ik}\}$\n\n, | x:$\{x_{ik}\}$ f(x):$\{y_{ik}\}$\n, |
| x:$\{x_{ik}\}$\|y:$\{y_{ik}\}$\n\n, | A:$\{x_{ik}\}$ B:$\{y_{ik}\}$, |
| text:$\{x_{ik}\}$\|label:$\{y_{ik}\}$\n, | x:$\{x_{ik}\}$\n y:$\{y_{ik}\}$\n\n |

# D RESULTS INCLUDING INCORRECT ICL

For simplicity of presentation in Section 3, we filter the test set to cases where the model correctly predicts $y_q$ given a 10-shot ICL prompt containing query $x_q$. In this section we compare those results to the setting in which correct-answer filtering is *not* applied. When filtering is not applied, the causal effects of function vectors remain essentially unchanged (Figure 7).

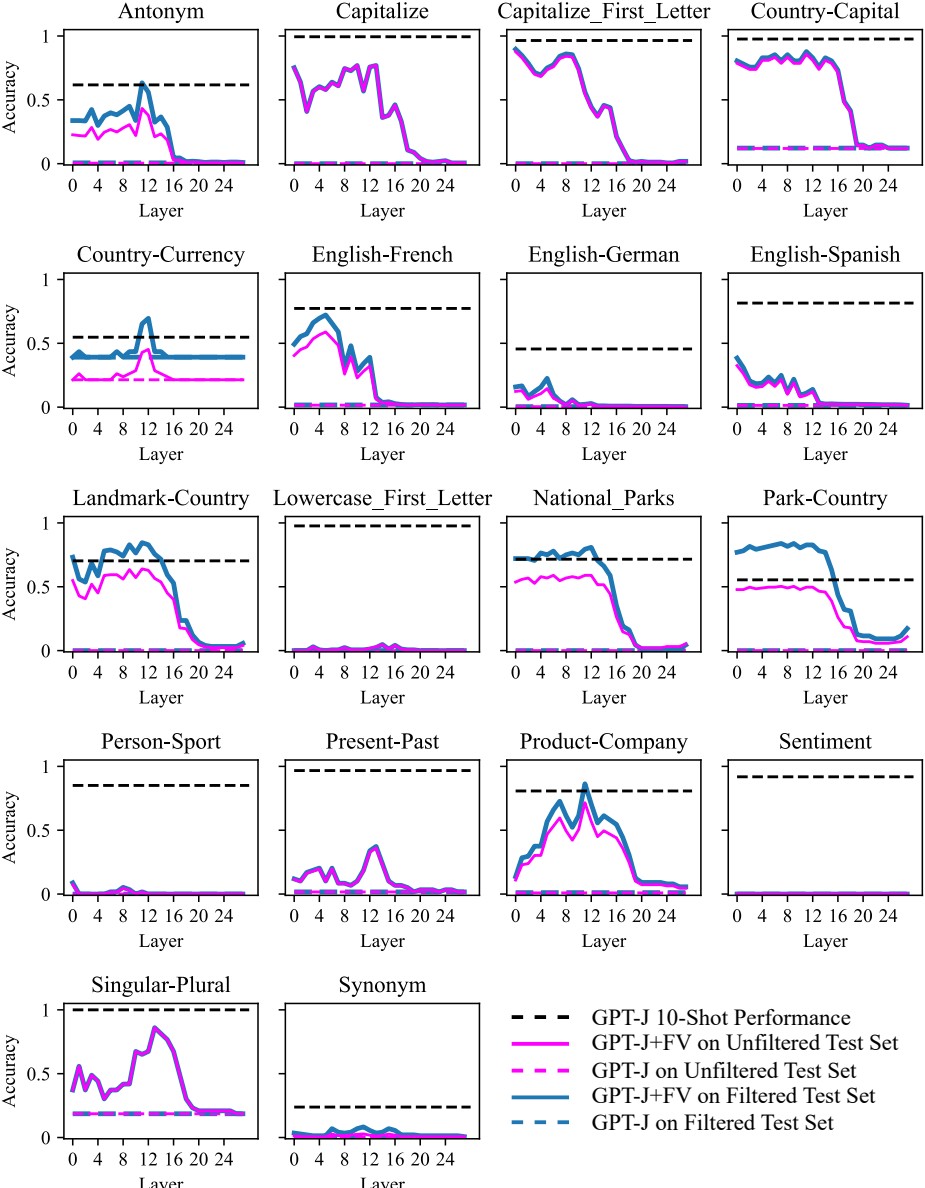

Figure 7: Comparing layer-wise zero-shot results of adding a function vector to GPT-J with and without filtering the task test set to cases where GPT-J correctly answers a 10-shot prompt. The results when filtering the test set in this manner are shown in blue, while the results without filtering the test set are shown in magenta. In both cases the performance is very similar, with performance dropping only slightly on a few tasks when not filtering the dataset to correct answers. The black dashed line corresponds to the oracle accuracy (GPT-J's 10-shot performance) in the unfiltered setting, while colored dashed lines correspond to GPT-J's performance in the zero-shot setting.

# E   DATASETS

Here, we describe the tasks we use for evaluating the existence of function vectors. A summary of each task can be found in Table 9.

**Antonym and Synonym.**   Our antonym and synonym datasets are based on data taken from Nguyen et al. (2017). They contain pairs of words that are either antonyms or synonyms of each other (e.g. "good → bad", or "spirited → fiery"). We create an initial dataset by combining all adjective, noun, and verb pairs from all data splits and then filter out duplicate entries. We then further filter to word pairs where both words can be tokenized as a single token. As a result, we keep 2,398 antonym word pairs and 2,881 synonym word pairs.

We note that these datasets originally included multiple entries for a single input word (e.g. both "simple → difficult" and "simple → complex" are entries in the antonym dataset). In those cases we prompt a more powerful model (GPT-4; OpenAI, 2023) with 10 ICL examples and keep an answer as output after manually verifying it.

**Translation.**   We construct our language translation datasets – English-French, English-German, and English-Spanish – using data from Conneau et al. (2017), which consists of a word in English and its translation into a target language. For each language, we combine the provided train and test splits into a single dataset and then filter out cognates. What remains are 4,705 pairs for English-French, 5,154 pairs for English-German, and 5,200 pairs for English-Spanish.

These datasets originally included multiple entries for a single input word (e.g. both "answer → respuesta" and "answer → contestar" are entries in the English-Spanish dataset), and so we filter those with GPT-4, in a similar manner as described for Antonym and Synonym.

**Sentiment Analysis.**   Our sentiment analysis dataset is derived from the Stanford Sentiment Treebank (SST-2) Socher et al. (2013), a dataset of movie review sentences where each review has a binary label of either "positive" or "negative". An example entry from this dataset looks like this: "An extremely unpleasant film. → negative". We use the same subset of SST-2 as curated in Honovich et al. (2023), where incomplete sentences and sentences with more than 10 words are discarded, leaving 1167 entries in the dataset. See Honovich et al. (2023) for more details.

**CommonsenseQA.**   This is a question answering dataset where a model is given a question and 5 options, each labeled with a letter. The model must generate the letter of the correct answer. For example, given "Where is a business restaurant likely to be located?" and answer options "a: town, b: hotel, c: mall, d: business sector, e: yellow pages", a model must generate "d". (Talmor et al., 2019)

**AG News.**   A text classification dataset where inputs are news headlines and the first few sentences of the article, and the labels are the category of the news article. Labels include Business, Science/Technology, Sports, and World. (Zhang et al., 2015)

We also construct a set of simple tasks to broaden the types of tasks on which we evaluate FVs.

**Capitalize First Letter.**   To generate a list of words to use to capitalize, we utilize ChatGPT[2] by prompting it to give us a list of words. From here, we curate a dataset where the input is a single word, and the output is the same word with the first letter capitalized.

**Lowercase First Letter.**   Similar to the Capitalize First Letter task, we use the same set of words but instead change the task to instead lowercase a word. The input is a single word title cased, and the output is the same word, but lowercase instead.

**Country-Capital.**   We also generate a list of country-capitals with ChatGPT. Here, we ask ChatGPT to come up with a list of countries and following that, ask it to name the capitals that are related to the countries. This dataset contains 197 country-capital city pairs.

---

[2] https://chat.openai.com/

**Country-Currency.**    We also generate a list of country-currency pairs with ChatGPT. Similar to country-capital, we ask ChatGPT to come up with a list of countries and following that, ask it to name the currencies that are related to the countries.

**National Parks.**    The National Parks dataset consists of names of official units of national parks in the United States, paired with the state the unit resides in (e.g. Zion National Park → Utah). It was collected from the corresponding page on Wikipedia[3], accessed July 2023.

**Park-Country.**    The Park-Country dataset consists of names of national parks around the world, paired with the country that the park is in. The countries were first generated by ChatGPT, then the parks were also generated by ChatGPT. After, the dataset was hand-checked for factual accuracy since ChatGPT tended to hallucinate parks for this dataset. A subset of all national parks is used.

**Present-Past.**    We generate a list of present-tense verbs with ChatGPT then ask ChatGPT to find the past-tense version. After generation, the dataset was hand-corrected for inaccuracies. The dataset inputs are simple present tense verbs and outputs are corresponding simple past tense verbs.

**Landmark-Country.**    The Landmark-Country dataset consists of entries with the name of a landmark, and the country that it is located in. The data pairs are taken from Hernandez et al. (2023b).

**Person-Instrument.**    The Person-Instrument dataset contains entries with the name of a professional musician and the instrument they play. The data pairs are taken from Hernandez et al. (2023b).

**Person-Occupation.**    The Person-Occupation dataset is taken from Hernandez et al. (2023b), and contains entries of names of well-known individuals and their occupations.

**Person-Sport.**    The Person-Sport dataset is taken from Hernandez et al. (2023b), and each entry consists of the name of a professional athlete and the sport that they play.

**Product-Company.**    The Product-Company dataset contains entries with the name of a commercial product, paired with the company that sells the product. It is curated from Hernandez et al. (2023b).

**Next-Item.**    The Next-Item dataset contains pairs of words which communicate the abstract idea of "next". Our pairs are made up of days of the week, months of the year, letters of the alphabet (which are cyclic), and number pairs (both numeric and text form). Some examples entries in this dataset are: "Monday" → "Tuesday", "December" → "January", "a" → "b", and "seven" → "eight".

**Previous-Item.**    The Previous-Item dataset contains the reciprocal version of the pairs of words in the "Next-Item" dataset, communicating the idea of "previous". Example entries include: "Tuesday" → "Monday", "January" → "December", and "a" → "z".

### E.1    EXTRACTIVE TASKS.

Many NLP tasks are **abstractive**; that is, they require the generation of information not present in the prompt. We also wish to test whether function vectors are recoverable from **extractive** tasks—that is, tasks where the answer is present somewhere in the prompt, and the task is to retrieve it.

**CoNLL-2003.**    In our experiments, we use a subset of the CoNLL-2003 English named entity recognition (NER) dataset Sang & De Meulder (2003), which is a common NLP benchmark for evaluating NER models. The NER task consists of extracting the correct entity from a given sentence, where the entity has some particular property. In our case, we create three different datasets: NER-person, NER-location, and NER-organization, where the label of each task is the name of either a person, location, or organization, respectively. Each dataset is constructed by first combining the CoNLL-2003 "train" and "validation" splits into a single dataset, and then filtering the data points to only include sentences where a single instance of the specified class (person, location, or organization) is present. This helps reduce the ambiguity of the task, as cases where multiple instances of the same class are present could potentially have multiple correct answers.

---

[3]`https://en.wikipedia.org/wiki/List_of_the_United_States_National_Park_System_official_units`

As with abstractive tasks, we also construct a set of new extractive tasks.

**Choose $n$th Item from List.** Here, the model is given a list of comma-separated items, and the model is tasked with selecting the item at a specific index. We construct tasks where the list size is either 3 or 5. In our tasks, we have the model choose either the first element or last element in the list.

**Choose Category from List.** These tasks are similar to our choose $n$th element tasks, but instead, the model must select an item of a particular type within a list of 3 or 5 items. A word with the correct type is included once in the list while the remaining words are drawn from another category. The categories we test include the following: fruit vs. animal, object vs. concept, verb vs. adjective, color vs. animal, and animal vs. object.

| Task Name | Task Source |
|---|---|
| *Abstractive Tasks* | |
| Antonym | Nguyen et al. (2017) |
| Capitalize first letter | |
| Capitalize | |
| Country-capital | |
| Country-currency | |
| English-French | Conneau et al. (2017) |
| English-German | Conneau et al. (2017) |
| English-Spanish | Conneau et al. (2017) |
| Landmark-Country | Hernandez et al. (2023b) |
| Lowercase first letter | |
| National parks | |
| Next-item | |
| Previous-item | |
| Park-country | |
| Person-instrument | Hernandez et al. (2023b) |
| Person-occupation | Hernandez et al. (2023b) |
| Person-sport | Hernandez et al. (2023b) |
| Present-past | |
| Product-company | Hernandez et al. (2023b) |
| Singular-plural | |
| Synonym | Nguyen et al. (2017) |
| CommonsenseQA (MC-QA) | Talmor et al. (2019) |
| Sentiment analysis (SST-2) | Socher et al. (2013) |
| AG News | Zhang et al. (2015) |
| *Extractive Tasks* | |
| Adjective vs. verb | |
| Animal vs. object | |
| Choose first of list | |
| Choose middle of list | |
| Choose last of list | |
| Color vs. animal | |
| Concept vs. object | |
| Fruit vs. animal | |
| Object vs. concept | |
| Verb vs. adjective | |
| CoNLL-2003, NER-person | Sang & De Meulder (2003) |
| CoNLL-2003, NER-location | Sang & De Meulder (2003) |
| CoNLL-2003, NER-organization | Sang & De Meulder (2003) |

Table 9: Summary of tasks used in this study. Tasks without sources are tasks we construct.

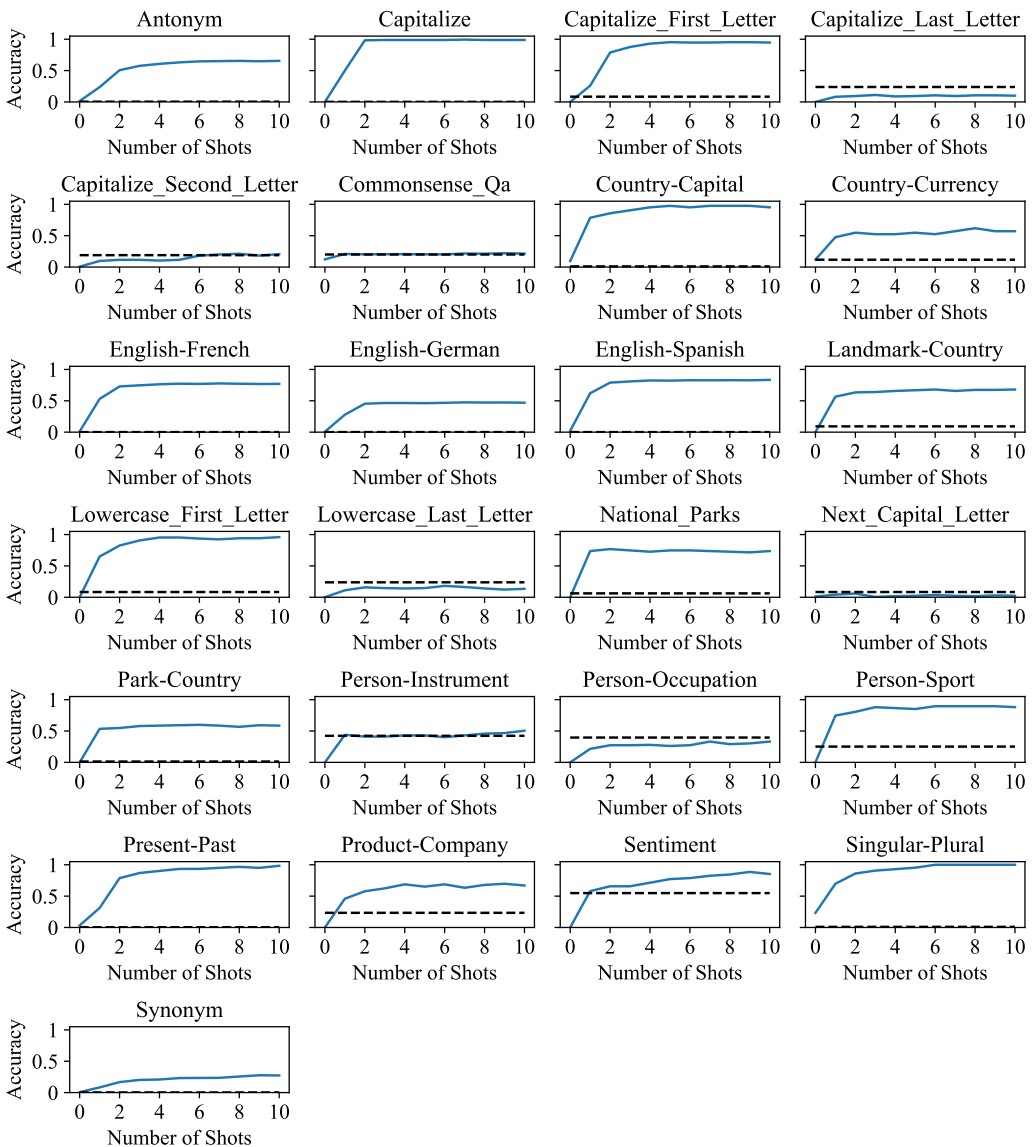

Figure 8: Few-shot ICL performance (top-1 accuracy) for GPT-J on a set of 25 abstractive-style tasks. In general, more shots improves performance. However, for many of these tasks the accuracy plateaus after a number of shots. The dotted baseline shows the accuracy of predicting only the majority label.

## E.2 FEW-SHOT ICL PERFORMANCE

Figure 8 shows the few-shot performance (top-1 accuracy) for GPT-J on a larger subset of our task list. The dotted baseline is based on the majority label for the dataset, computed as (# majority label)/(# total instances).

Figure 9 shows the few-shot performance (top-1 accuracy) for GPT-J on additional extractive tasks. For datasets with lists of words as input, the dotted baseline is computed to be $1/$size of the input list, (e.g. $1/3, 1/5$), and for all other tasks it represents predicting the majority label of the dataset, computed as (# majority label)/(# total instances).

## E.3 EVALUATING FUNCTION VECTORS ON ADDITIONAL TASKS

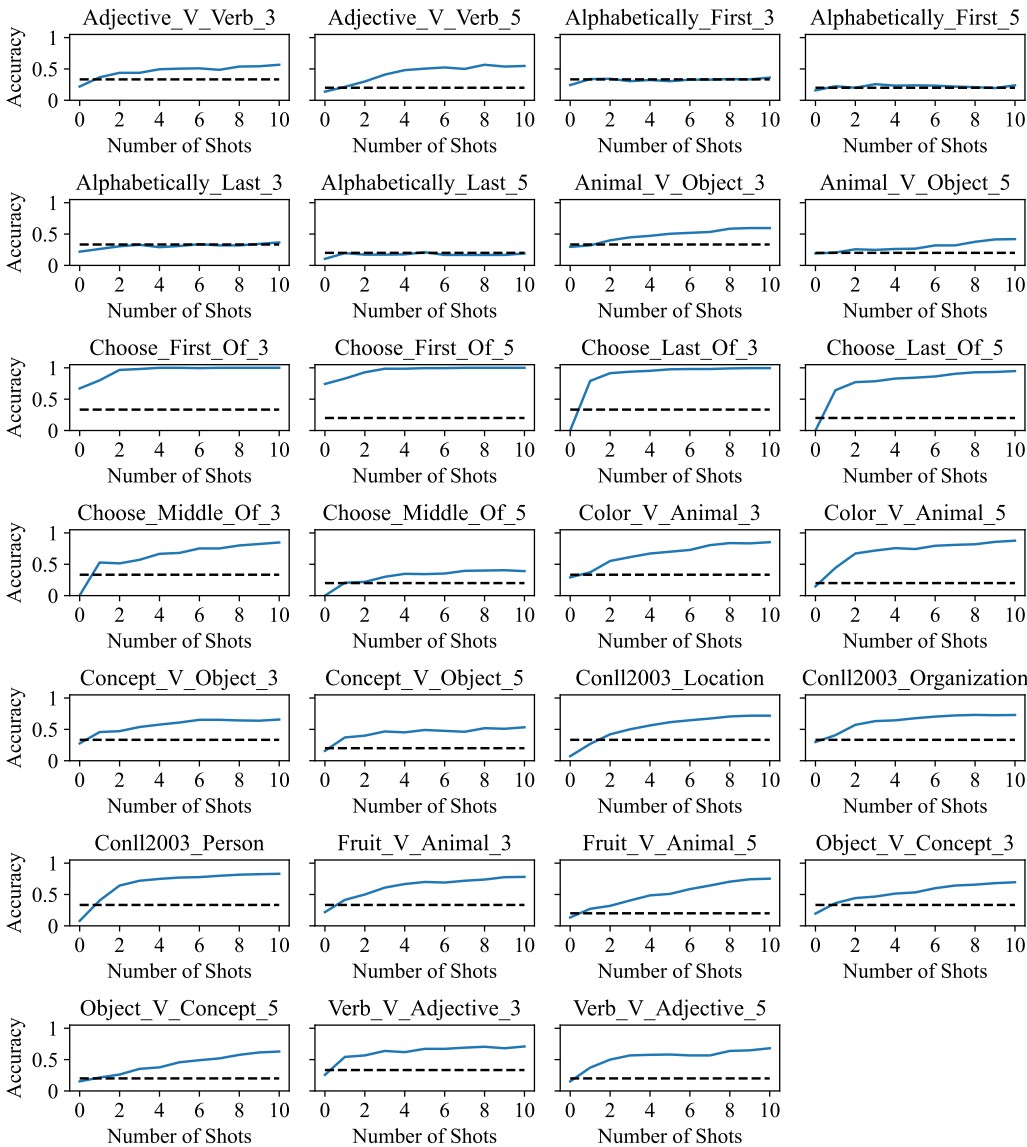

Figure 9: Few-shot ICL performance (top-1 accuracy) for GPT-J on a set of 27 extractive-style tasks. In general, more shots improves model performance. A dataset with 3 or 5 at the end denotes the size of the input list of words used. There are a few cases where the model cannot perform the task any better than random (e.g. choose the alphabetically first word in a list), which we do not analyze further. The dotted baseline shows the accuracy $1/(\text{list size})$, or the majority label in the case of the datasets derived from conll2003.

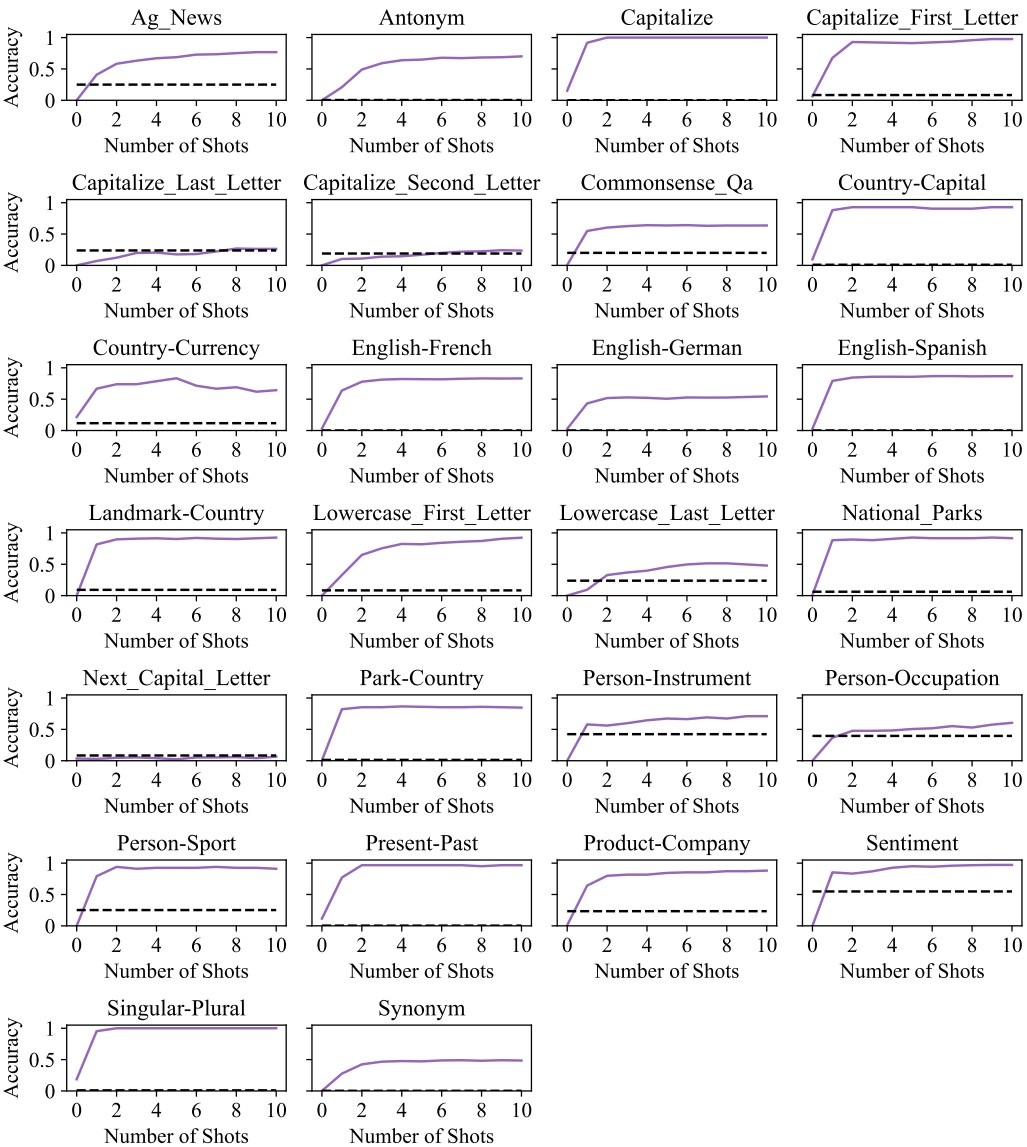

Figure 10: Few-shot ICL performance (top-1 accuracy) for Llama 2 (13B) on a set of 26 abstractive-style tasks. Using more ICL examples (shots) improves performance for many of these tasks, though the performance does plateau after a few shots for many of the tasks. There are a few cases where Llama 2 (13B) cannot perform the task any better than random (e.g. capitalize the second letter in the word), which we do not analyze further. The dotted baseline shows accuracy choosing the majority label.

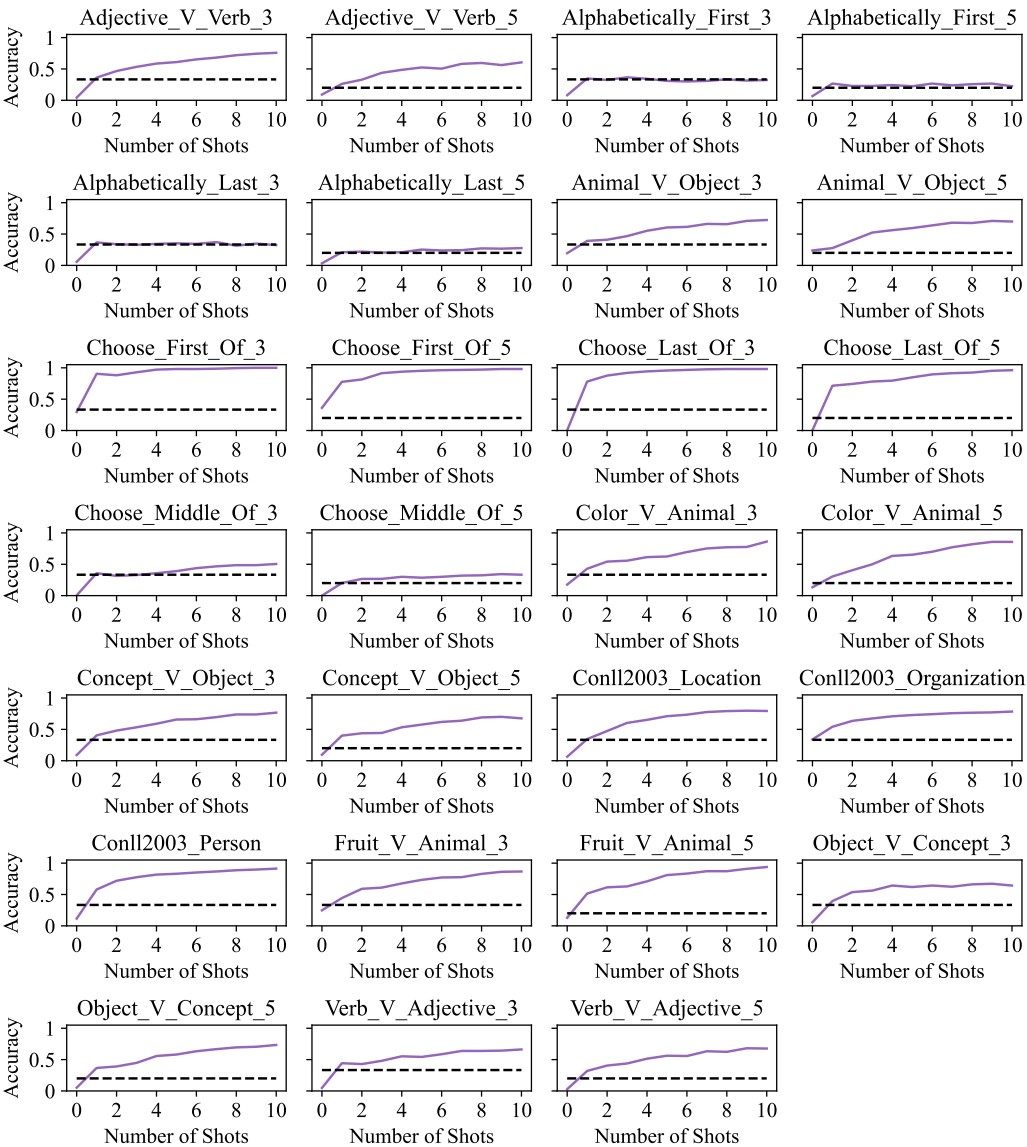

Figure 11: Few-shot ICL performance (top-1 accuracy) for Llama 2 (13B) on a set of 27 extractive-style tasks. Using more ICL examples improves performance for most of the tasks. A dataset name ending with 3 or 5 denotes the size of the input word list. There are a few cases where the model cannot perform the task any better than random (e.g. choose the alphabetically first word in a list), which we do not analyze further. The dotted baseline shows the accuracy $1/(\text{list size})$ (e.g. $1/3, 1/5$, or the majority label in the case of the conll2003 datasets).

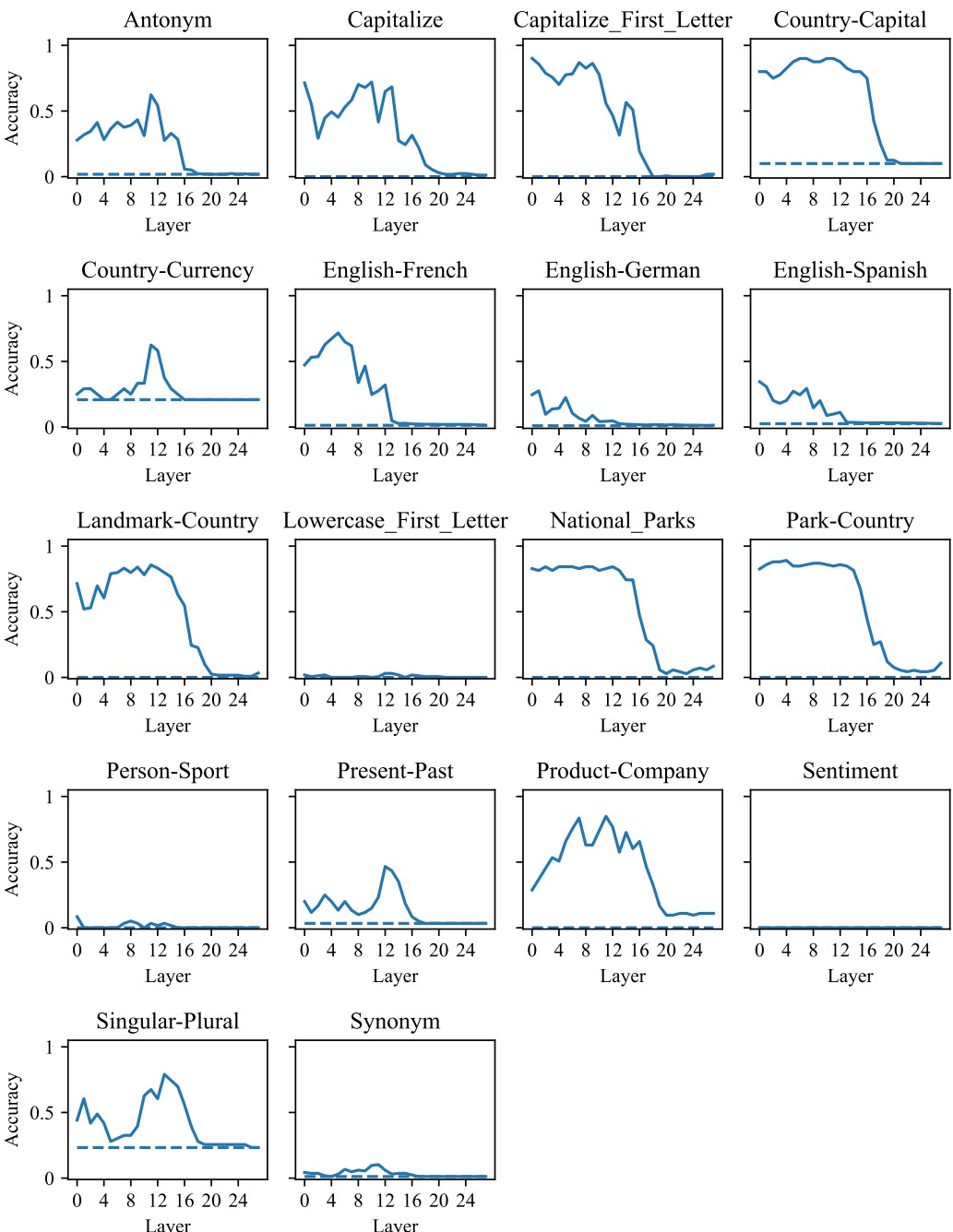

Figure 12: Zero-Shot Top-1 Accuracy Results of adding FVs to GPT-J across 18 of our task datasets. In addition the 6 analyzed in the main paper, we see similar results and performance across a variety of other tasks - adding FVs in early-middle layers seems to have the most effect. Notable exceptions include lowercase first letter, person-sport, synonym, and sentiment, where the FV doesn't seem to have much effect. Note we did not evaluate the FV on tasks where the model's ICL performance was poor (compared to the majority label baseline) (Figure 8).

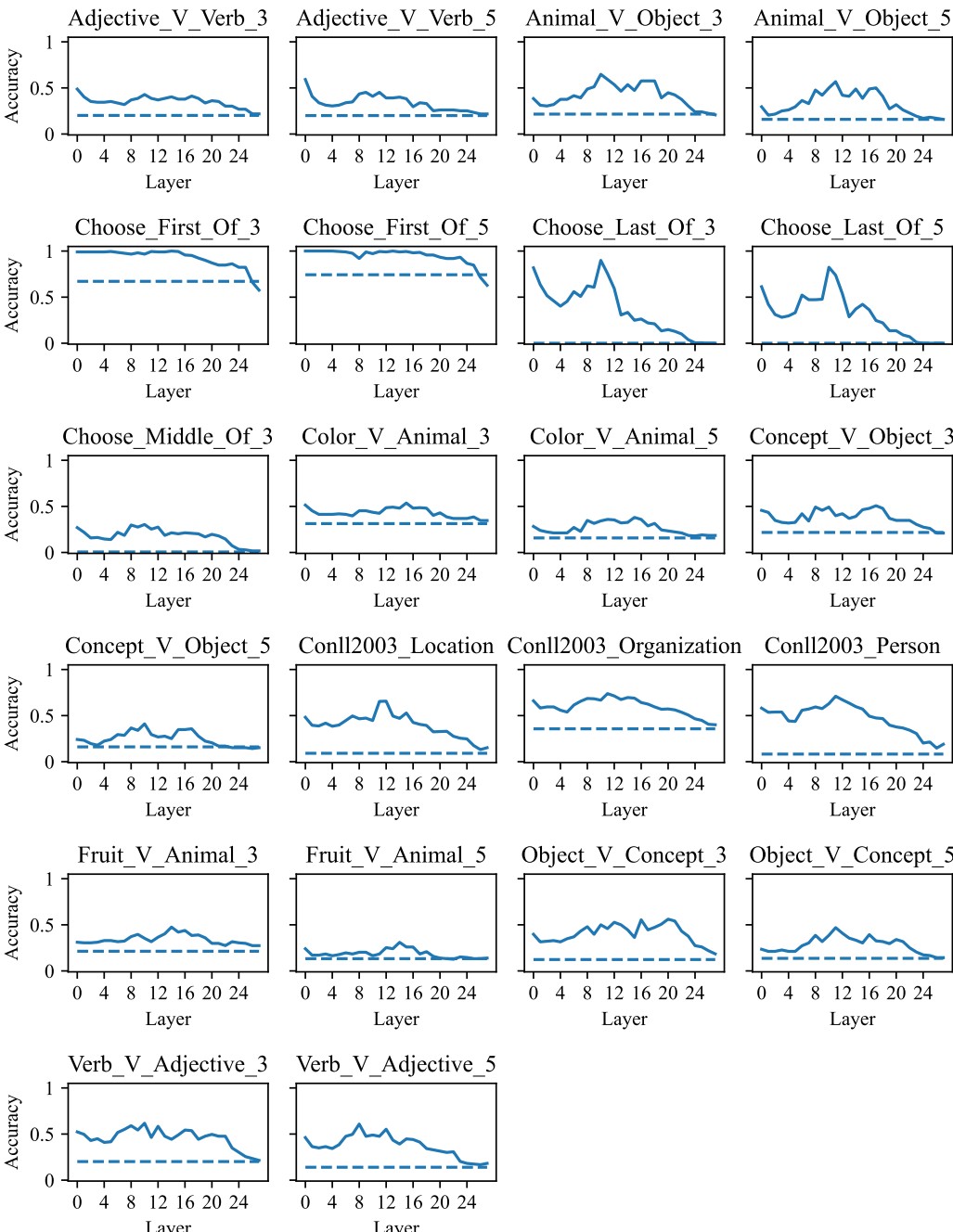

Figure 13: Zero-Shot Top-1 Accuracy Results of adding FVs to GPT-J across 22 of our task datasets. Shown here are mainly extractive tasks, and we see similar trends across all tasks. The FV performance is fairly stagnant across layers, with many tasks having peak performance in middle layers of the network. The model's zero-shot performance without intervention is plotted as a dotted line. Adding the FV causes the model to extract the correct entity much more often than the base model in this uninformative zero-shot case.

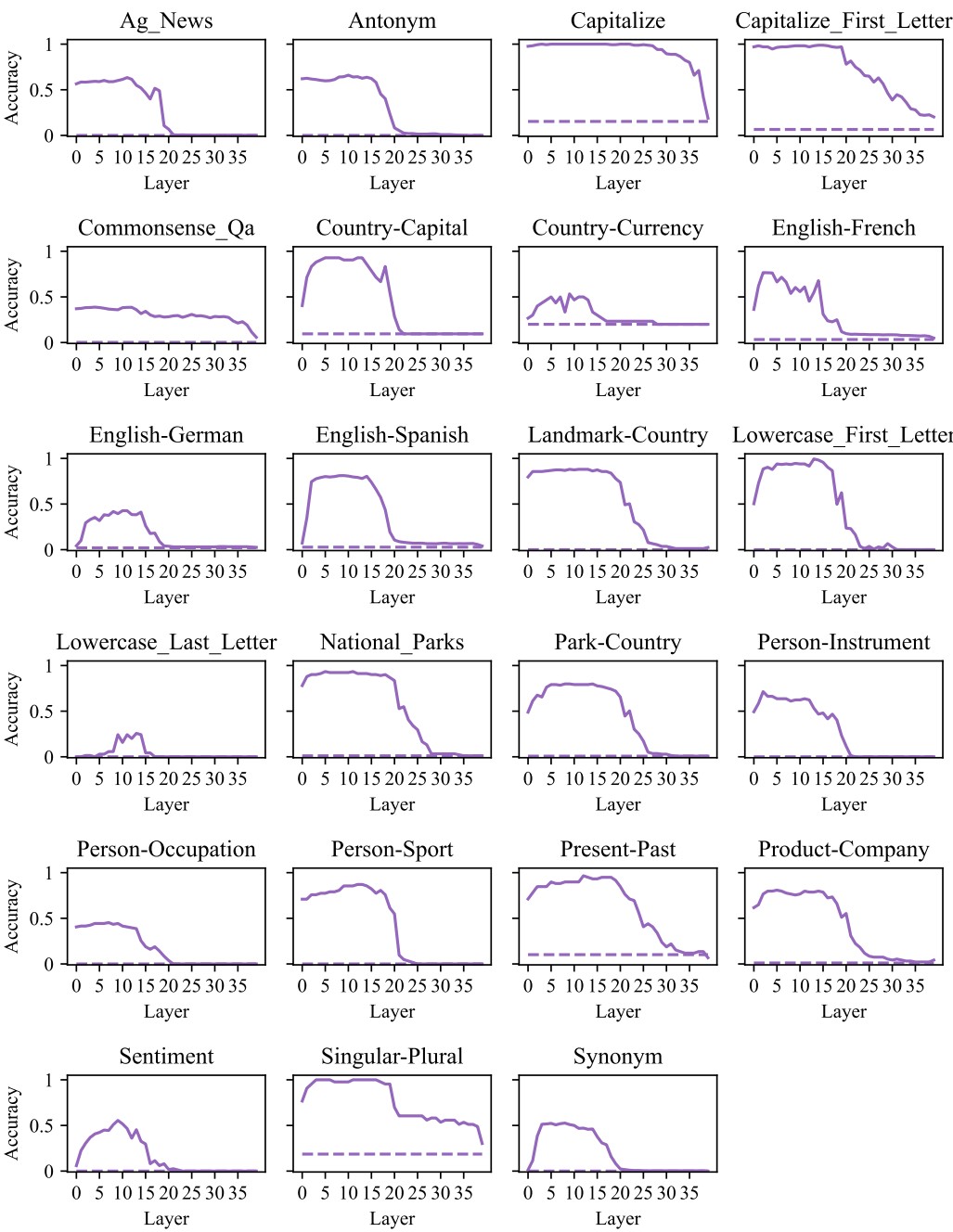

Figure 14: Zero-shot results for adding a function vector (FV) at different layers of Llama 2 (13B) across a 23 abstractive-style tasks. Adding the FV at early-middle layers gives good performance on the task, but there is a drop in performance when adding the FV at later layers of the model (around layer 20-25 for Llama 2 (13B)).

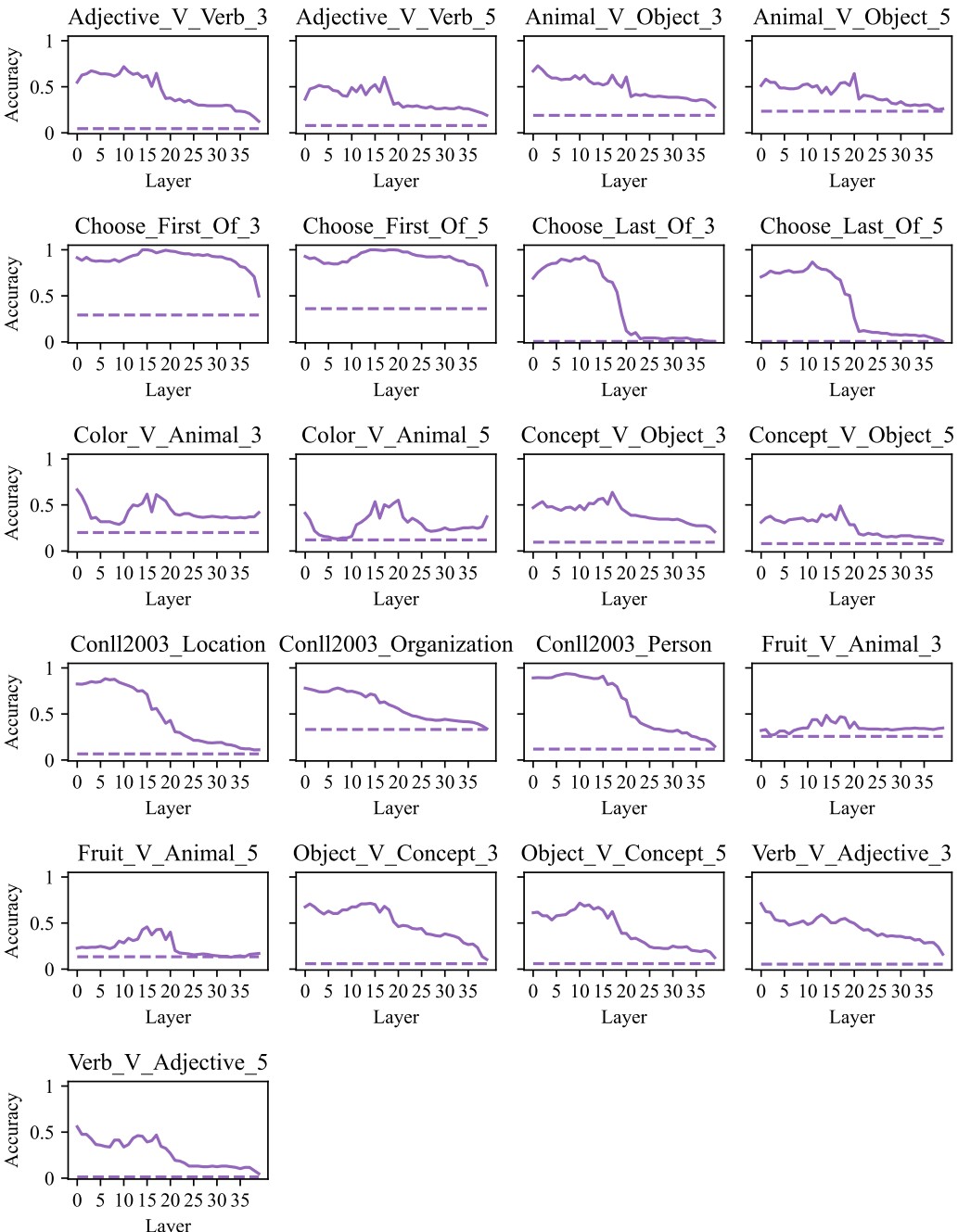

Figure 15: Zero-shot results for adding a function vector (FV) at different layers of Llama 2 (13B) across a 21 extractive-style tasks. Adding the FV at earlier layers has a higher causal effect of performing the extracted task, while later layers see a performance dip.

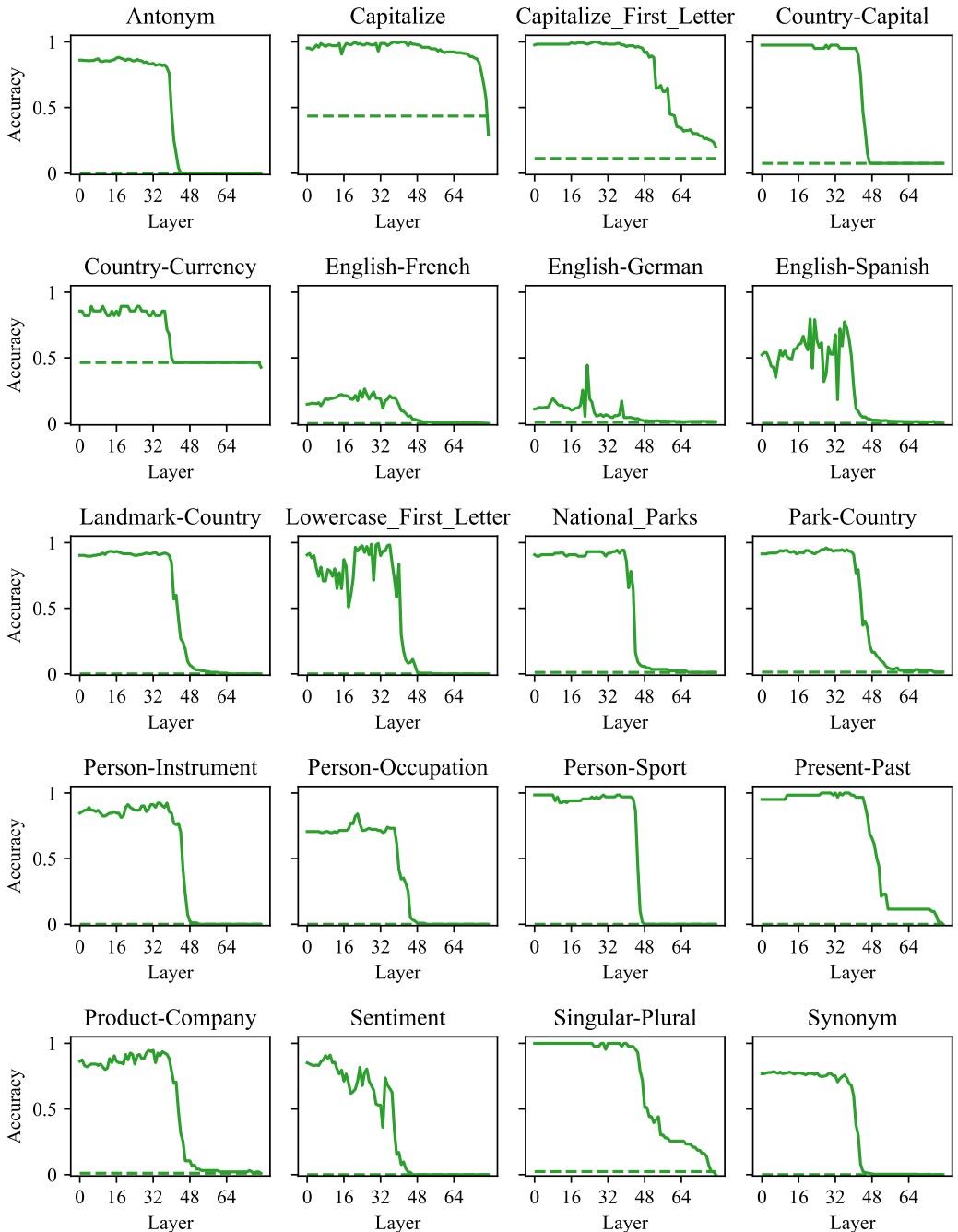

Figure 16: Zero-shot results for adding a function vector (FV) at different layers of Llama 2 (70B) across a 20 abstractive-style tasks. Adding the FV at early-middle layers gives good performance on the task, and there is a sharp drop in performance when adding the FV at later layers of the model (after layer 48 for Llama 2 (70B)).

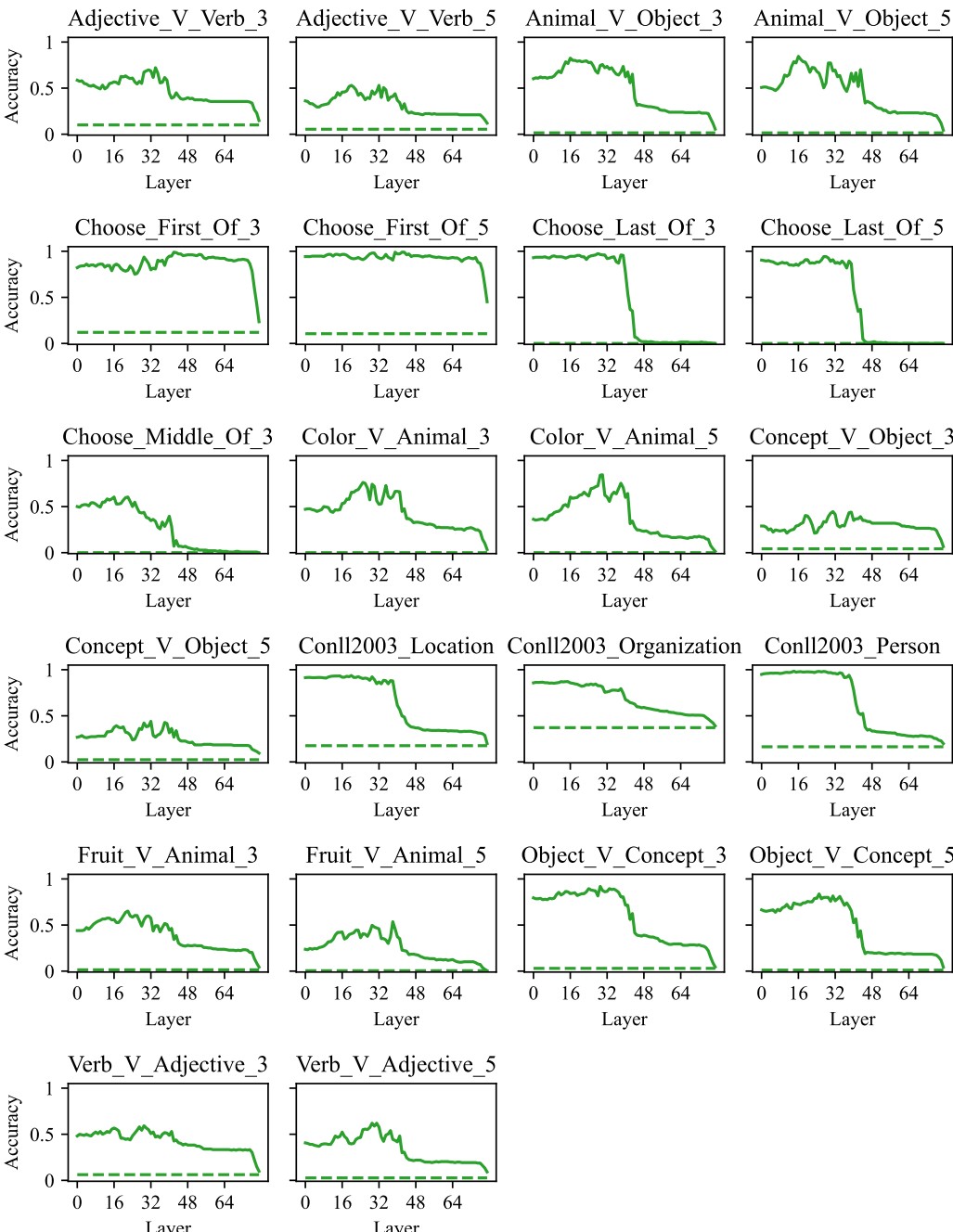

Figure 17: Zero-shot results for adding a function vector (FV) at different layers of Llama 2 (70B) across 22 extractive-style tasks. In general, adding the FV at early-middle layers gives good performance on the task, and there is often a sharp drop in performance when adding the FV at later layers of the model (around layer 48 for Llama 2 (70B)).

# F   PORTABILITY

In addition to the quantitative results provided in §3.1, we include additional qualitative examples for natural text completions for the Antonym FV (Table 10). We also include qualitative examples of completions on different natural text prompts using the English-French FV (Table 11), the English-Spanish FV (Table 13), and the Country-Capital FV (Table 14).

We also include additional quantitative results for testing FVs in natural text settings on various natural text templates, averaged over 5 seeds for each of the remaining representative tasks: Capitalize (Table 16), Present-Past (Table 17), Singular-Plural (Table 18), English-French (Table 12), and Country-Capital (Table 15).

Table 10: Additional Qualitative Examples of the Antonym FV in Naturalistic Text Settings

| **Antonym Task** |
| --- |
| **Prompt:** The word "{x}", means |
| **(a) GPT-J** The word "limitless", means "without limits".\n\nThe word
**(a) GPT-J + Antonym FV** The word "limitless", means "finite".\n\nThe word |
| **(b) GPT-J** The word "improvement", means "to make better".\n\nThe
**(b) GPT-J + Antonym FV** The word "improvement", means "deterioration".\n\n |
| **Prompt:** When I think of {x}, I usually |
| **(a) GPT-J** When I think of sit, I usually think of a chair. But, in"
**(a) GPT-J + Antonym FV** When I think of sit, I usually think of stand.\n\n |
| **(b) GPT-J** When I think of maximum, think of the maximum number of people that
**(b) GPT-J + Antonym FV** When I think of maximum, I usually think of minimum.\n\nI think |
| **Prompt:** While reading a book, I came across the word "{x}". I looked it up in a dictionary and it turns out that it means |
| **(a) GPT-J** While reading a book, I came across the word "credible". I looked it up in a dictionary and it turns out that it means "believable".\n\nI
**(a) GPT-J + Antonym FV** While reading a book, I came across the word "credible". I looked it up in a dictionary and it turns out that it means "unbelievable".\n\n |
| **(b) GPT-J** While reading a book, I came across the word "marriage". I looked it up in a dictionary and it turns out that it means "a legal union between a man and
**(b) GPT-J + Antonym FV** While reading a book, I came across the word "marriage". I looked it up in a dictionary and it turns out that it means "divorce".\n\nI came |
| **Prompt:** The word {x} can be understood as a synonym for |
| **(a) GPT-J** The word erroneous can be understood as a synonym for wrong, but it is also a synonym for
**(a) GPT-J + Antonym FV** The word erroneous can be understood as a synonym for correct.\n\nThe |
| **(b) GPT-J** The word injure can be understood as a synonym for harm, but it also has a more specific meaning
**(b) GPT-J + Antonym FV** The word injure can be understood as a synonym for heal.\n\nThe word |

Table 11: Evaluating the English-French FV in Naturalistic Text Settings

| English-French Task |
|---|
| **Prompt:** The word "{x}", means |
| **(a) GPT-J** The word "readers", means "people who read".\n\nThe
**(a) GPT-J + English-French FV** The word "readers", means "lecteurs"\n\nThe |
| **(b) GPT-J** The word "sky", means "the heavens" in many languages.
**(b) GPT-J + English-French FV** The word "sky", means "ciel" in français |
| **(c) GPT-J** The word "daily", means "every day".\n\nThe word
**(c) GPT-J + English-French FV** The word "daily", means "tous les jours", |
| **Prompt:** The word '{x}' can be understood as a synonym for |
| **(a) GPT-J** The word 'link' can be understood as a synonym for 'connection' or 'relation'. The term 'link' is used in the context of the Internet, but it can be used in other contexts as
**(a) GPT-J + English-French FV** The word 'link' can be understood as a synonym for 'lien', et le mot 'lien' peut être compris comme un synonyme de 'lien', et |
| **(b) GPT-J:** The word 'prohibited' can be understood as a synonym for 'forbidden' or 'forbidden to do'.
**(b) GPT-J + English-French FV** "The word 'prohibited' can be understood as a synonym for 'interdit' |
| **Prompt:** While reading a book, I came across the word "{x}". I looked it up in a dictionary and it turns out that it means |
| **(a) GPT-J** While reading a book, I came across the word "knight". I looked it up in a dictionary and it turns out that it means "a person who is a member of a military order".\n\nI was wondering if there is a similar word for a person who is a member
**(a) GPT-J + English-French FV** While reading a book, I came across the word "knight". I looked it up in a dictionary and it turns out that it means "chevalier".\n\nJe lis un livre, et cést un chevalier. |

Table 12: Natural text portability of the English-French FV. Given a natural template we substitute a query word for 'x'. We measure accuracy based on whether the correct french translation is produced in this setting within 5 generated tokens.

| Prompt | GPT-J | +English-French FV |
|---|---|---|
| The word "x", means | $0.0 \pm 0.0\%$ | $51.2 \pm 1.2\%$ |
| When I think of the word "x", it usually means | $0.1 \pm 0.1\%$ | $41.7 \pm 1.3\%$ |
| When I think of x, I usually | $0.0 \pm 0.0\%$ | $19.0 \pm 1.5\%$ |
| While reading a book, I came across the word "x". I looked it up in a dictionary and it turns out that it means | $0.1 \pm 0.1\%$ | $45.6 \pm 1.1\%$ |
| The word x can be understood as a synonym for | $0.3 \pm 0.2\%$ | $44.2 \pm 1.6\%$ |

Table 13: Evaluating the English-Spanish FV in Naturalistic Text Settings

| English-Spanish Task |
|---|
| **Prompt:** The word "{x}", means

**(a) GPT-J** The word "children", means "offspring of the human species".\n \nThe word "children" is used in the Bible
**(a) GPT-J + English-Spanish FV** The word "children", means "niños" in español.\n\nThe word "children", means

**(b) GPT-J** The word "foods", means "food" in the sense of "food" and "drink" in the sense of
**(b) GPT-J + English-Spanish FV** The word "foods", means "alimentos"\n\nThe word "alimentos", significa "alimentos |
| **Prompt:** While reading a book, I came across the word "{x}". I looked it up in a dictionary and it turns out that it means

**(a) GPT-J** While reading a book, I came across the word "attacking". I looked it up in a dictionary and it turns out that it means "to attack".\n\nI was wondering if there is a difference between the two words.
**(a) GPT-J + English-Spanish FV** While reading a book, I came across the word "attacking". I looked it up in a dictionary and it turns out that it means "atacar".\n\nI leído un libro y encontré la palabra "atacar".\n\nPero

**(b) GPT-J** While reading a book, I came across the word "computer". I looked it up in a dictionary and it turns out that it means "a machine that computes".\n\nI was surprised to learn that the word "computer" is derived from the word "computor",
**(b) GPT-J + English-Spanish FV** While reading a book, I came across the word "computer". I looked it up in a dictionary and it turns out that it means computador.\n\nI looked it up in a dictionary y en un diccionario y que dice que es un computador.\n\n |
| **Prompt:** When I think of {x}, I usually

**(a) GPT-J** When I think of find, I usually think of the command line. It's
**(a) GPT-J + English-Spanish FV** When I think of find, I usually think de encontrar.\n\nWhen I

**(b) GPT-J** When I think of tomb, I usually think of a tombstone. But, in the case of the tomb of
**(b) GPT-J + English-Spanish FV** When I think of tomb, I usually pense en cámara de sepulcro.\n\nCuando

**(c) GPT-J** When I think of shoes, I usually think of the ones I wear. I'm not a shoe person
**(c) GPT-J + English-Spanish FV** When I think of shoes, I usually pense sobre zapatos.\n\nWhen I pense sobre |

Table 14: Evaluating the Country-Capital FV in Naturalistic Text Settings

| Country-Capital Task |
|---|
| **Prompt:** A couple of years ago I visited {x}, and |
| **(a) GPT-J:** A couple years ago I visited Grenada, and I was struck by the beauty of the island. I was also struck by the fact that the island was a British colony until 1974. I was also
**(a) GPT-J + Country-Capital FV** A couple years ago I visited Grenada, and I was in the capital, St. George's. I was there to see the Grenada National Museum, and I was there to see |
| **(b) GPT-J:** A couple years ago I visited Italy, and I was lucky enough to be able to visit the city of Florence. I was there for a few days, and I was able to see a lot
**(b) GPT-J + Country-Capital FV** A couple years ago I visited Italy, and I was in Rome. I was in the Colosseum, and I was in the Roman Forum. I was in the Pantheon, and |
| **(c) GPT-J:** couple years ago I visited Belarus, and I was surprised to see that the country was still using the old Soviet-era typewriters. I was even more surprised to see that the typew
**(c) GPT-J + Country-Capital FV** A couple years ago I visited Belarus, and I was in Minsk. I was there for a few days, and I was in Minsk |
| **(d) GPT-J:** A couple years ago I visited Mongolia, and I was struck by the fact that the country was so different from the rest of the world. It was a land of nomads, and the people
**(d) GPT-J + Country-Capital FV** A couple years ago I visited Mongolia, and I was in Ulaanbaatar. I was there for a few days, and I was in the capital city of Ulaanbaatar |
| **Prompt:** If you ever travel to {x}, you have to visit |
| **(a) GPT-J:** If you ever travel to Italy, you have to visit the city of Florence. It is the capital of Tuscany and the most visited city in Italy. It is also the home of the famous U
**(a) GPT-J + Country-Capital FV** If you ever travel to Italy, you have to visit Rome. It's the capital of Italy and the largest city in the world. It's also the most visited city in the world |
| **(b) GPT-J:** If you ever travel to Thailand, you have to visit the island of Koh Samui. It is a beautiful island with a lot of things to do. The island is famous for its beaches, water sports
**(b) GPT-J + Country-Capital FV** If you ever travel to Thailand, you have to visit Bangkok. It is the capital of Thailand and the largest city in the country. Bangkok is the most populous city in the world.\n\nBangkok |
| **(c) GPT-J:** If you ever travel to Saint Lucia, you have to visit the Pitons. The Pitons are a group of three mountains that are located on the island of Saint Lucia. The Pitons are the highest mountains
**(c) GPT-J + Country-Capital FV** If you ever travel to Saint Lucia, you have to visit the capital city of Castries. It is the most beautiful city in the Caribbean. It is a very beautiful city. It is a very beautiful city |
| **Prompt:** When you think of {x}, |
| **(a) GPT-J:** When you think of Netherlands, you probably think of tulips, windmills, and cheese. But the Netherlands is also home to a thriving cannabis industry
**(a) GPT-J + Country-Capital FV** When you think of Netherlands, you think of Amsterdam. But there are many other cities in the Netherlands. Here are some of the best places to visit in |
| **(b) GPT-J:** When you think of Egypt, you probably think of pyramids, mummies, and the Nile River. But did you know that Egypt is also home to
**(b) GPT-J + Country-Capital FV** When you think of Egypt, you think of Cairo, the pyramids, and the Nile. But there are many other places to visit in Egypt. Here |

Table 15: Natural Text Portability quantitative results for the Country-Capital FV. We substitute in query country names for 'x', and measure accuracy based on whether the correct capital city name is produced within 10 generated tokens.

| Prompt | GPT-J | +Country-Capital FV |
|---|---|---|
| A couple of years ago I visited {x}, and | $3.9 \pm 3.2\%$ | $56.7 \pm 5.3\%$ |
| If you ever travel to {x}, you have to visit | $23.2 \pm 3.5\%$ | $70.4 \pm 3.7\%$ |
| When you think of {x}, | $7.4 \pm 4.8\%$ | $72.4 \pm 2.7\%$ |

Table 16: Natural text portability of the Capitalize FV. Given a natural template we substitute a query word for 'x'. We measure accuracy based on whether the correct capitalization is produced in this natural text setting within 5 generated tokens.

| Prompt | GPT-J | +Capitalize FV |
|---|---|---|
| The word "x", means | $5.6 \pm 0.5\%$ | $94.3 \pm 1.0\%$ |
| When I think of the word "x", it usually means | $3.7 \pm 0.7\%$ | $84.5 \pm 1.3\%$ |
| When I think of x, I usually | $12.5 \pm 2.1\%$ | $76.1 \pm 2.9\%$ |
| While reading a book, I came across the word "x". I looked it up in a dictionary and it turns out that it means | $6.8 \pm 0.7\%$ | $97.8 \pm 0.6\%$ |
| The word x can be understood as a synonym for | $5.5 \pm 0.8\%$ | $81.5 \pm 2.8\%$ |

Table 17: Natural text portability of the Present-Past FV. Given a natural template we substitute a query word for 'x'. Then then measure accuracy based on whether under the FV intervention the correct past-tense word is produced within 5 generated tokens.

| Prompt | GPT-J | +Present-Past FV |
|---|---|---|
| The word "x", means | $0.0 \pm 0.0\%$ | $49.3 \pm 4.5\%$ |
| When I think of the word "x", it usually means | $0.1 \pm 0.3\%$ | $67.2 \pm 4.2\%$ |
| When I think of x, I usually | $0.7 \pm 0.7\%$ | $16.1 \pm 5.4\%$ |
| While reading a book, I came across the word "x". I looked it up in a dictionary and it turns out that it means | $0.0 \pm 0.0\%$ | $55.8 \pm 3.6\%$ |
| The word x can be understood as a synonym for | $0.0 \pm 0.0\%$ | $57.2 \pm 3.0\%$ |

Table 18: Natural text portability of the Singular-Plural FV. Given a natural template we substitute a query word for 'x'. Then then measure accuracy based on whether under the FV intervention the correct past-tense word is produced within 5 generated tokens.

| Prompt | GPT-J | +Singular-Plural FV |
|---|---|---|
| The word "x", means | $0.0 \pm 0.0\%$ | $82.9 \pm 3.6\%$ |
| When I think of the word "x", it usually means | $0.0 \pm 0.0\%$ | $72.2 \pm 0.6\%$ |
| When I think of x, I usually | $0.0 \pm 0.0\%$ | $36.0 \pm 8.4\%$ |
| While reading a book, I came across the word "x". I looked it up in a dictionary and it turns out that it means | $0.0 \pm 0.0\%$ | $81.1 \pm 2.4\%$ |
| The word x can be understood as a synonym for | $0.3 \pm 0.6\%$ | $77.0 \pm 5.0\%$ |

## G  CAUSAL MEDIATION ANALYSIS

In this section we include additional figures showing the average indirect effect (AIE) split up across tasks for GPT-J (Figure 18), as well as the AIE for other models we evaluated.

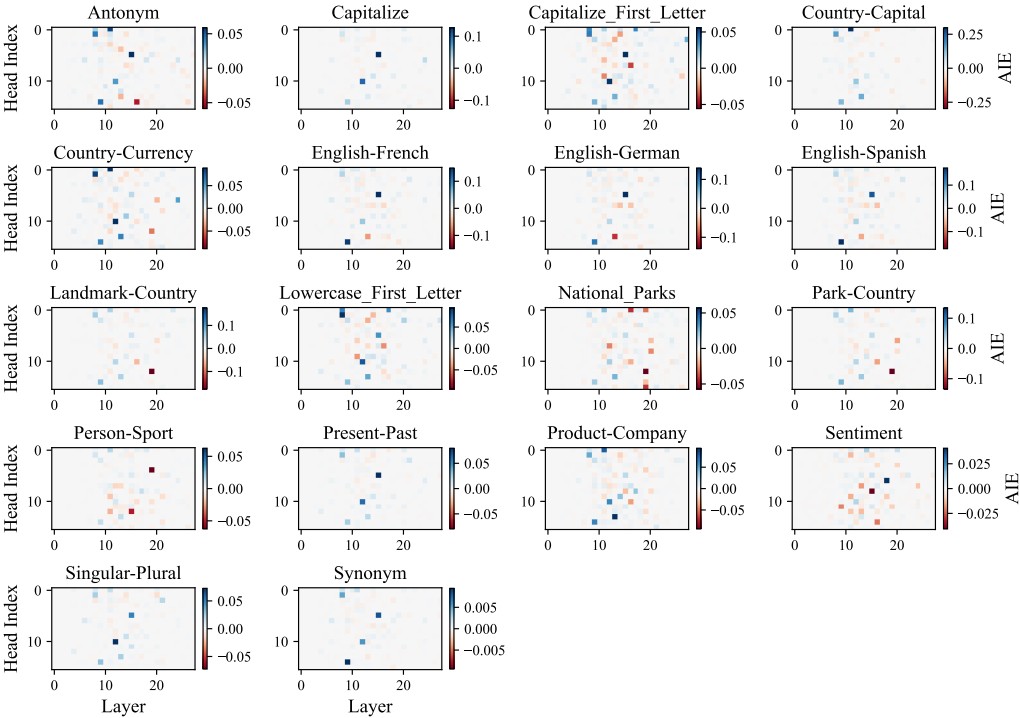

Figure 18: Average Indirect Effect (AIE) for attention heads in GPT-J, shown by task. The set of heads that have higher causal effect is fairly consistent across tasks.

The AIE for a particular model is computed across all abstractive tasks where the model can perform the task better than the majority label baseline given 10 ICL examples. We include heatmaps of the AIE for each attention head in Llama 2 (7B) (Figure 19), Llama 2 (13B) (Figure 20), and Llama 2 (70B) (Figure 22), as well as GPT-NeoX (Figure 21). For each model, the heads with the highest AIE are typically clustered in the middle layers of the network. In addition the maximum AIE across all heads tends to drop slightly as the size of the model increases, though the total number of heads also increases. In Llama 2 (7B) the max AIE is $\approx 0.047$, while in Llama 2 (70B) the max AIE is $\approx 0.037$.

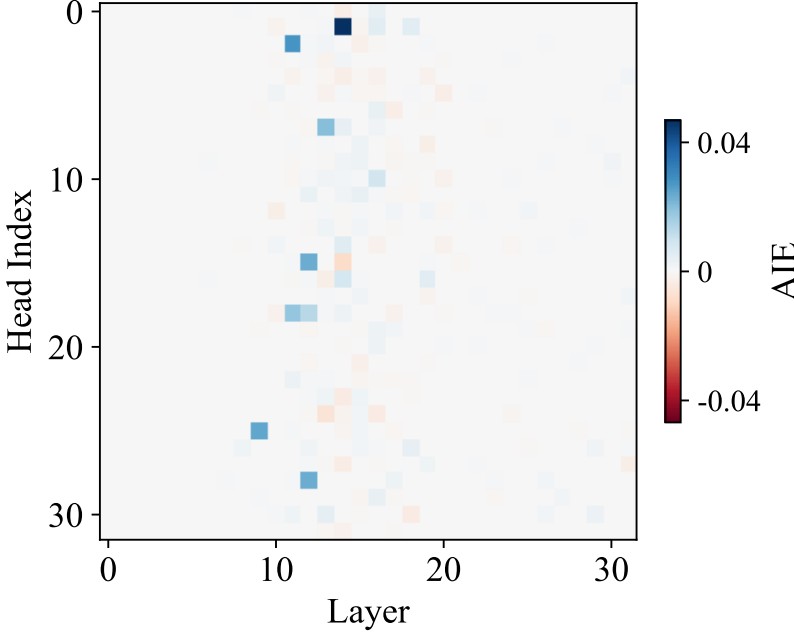

Figure 19: Average Indirect Effect (AIE) for each attention head in Llama 2 (7B) at the final token. This is computed across all abstractive tasks where the model can perform the task with 10 ICL examples. The heads with the highest AIE are mainly clustered in the middle layers of the network. Compared to GPT-J the AIE of the most influential heads is less ($\approx 0.047$ vs. $\approx 0.053$ in GPT-J) but there are also more than double the number of attention heads

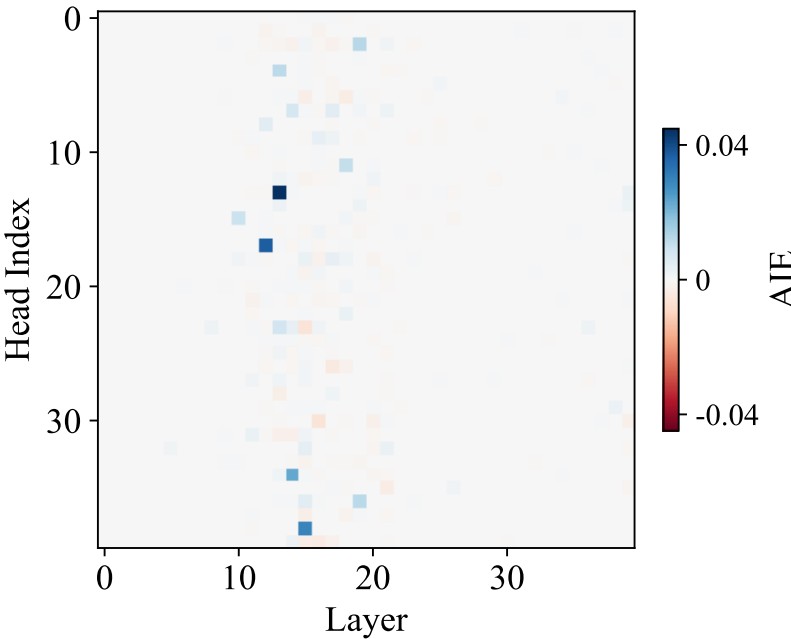

Figure 20: Average Indirect Effect (AIE) for each attention head in Llama 2 (13B) at the final token.

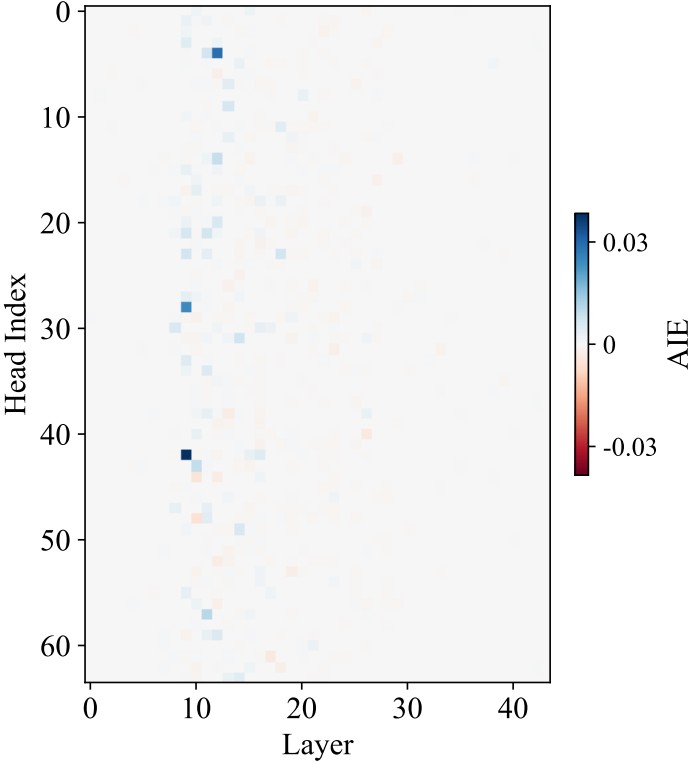

Figure 21: Average Indirect Effect (AIE) for each atttention head in GPT-NeoX at the final token. Interestingly, the heads with the highest AIE here are clustered in earlier middle layers (from layer 10-20), whereas in other models the heads are clustered more towards the middle of the network.

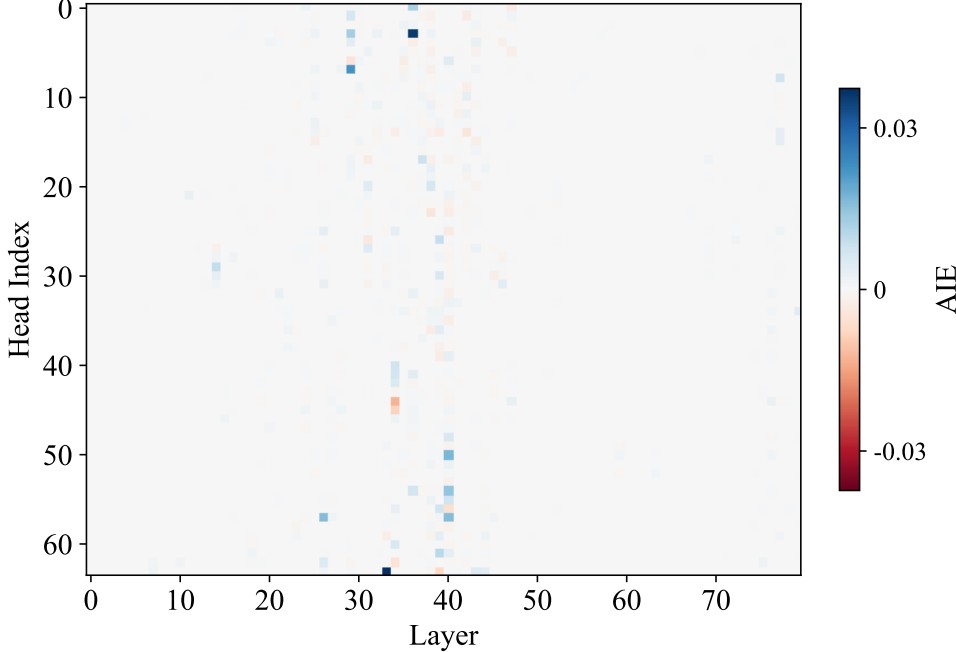

Figure 22: Average Indirect Effect (AIE) for each attention head in Llama 2 (70B) at the final token.

# H    ATTENTION PATTERNS AND PREFIX-MATCHING SCORE

Across a variety of tasks, the heads with highest causal effect have a consistent attention pattern where the attention weights on few-shot ICL prompts are the strongest on the output tokens of each in-context example. Here we show this pattern for GPT-J on 4 additional tasks (Figure 23, Figure 24), which match the patterns shown in the main paper (Figure 3b). This is similar to the attention pattern that might be expected of "induction heads", which has previously been shown to arise when a prompt contains some repeated structure (Elhage et al., 2021; Olsson et al., 2022).

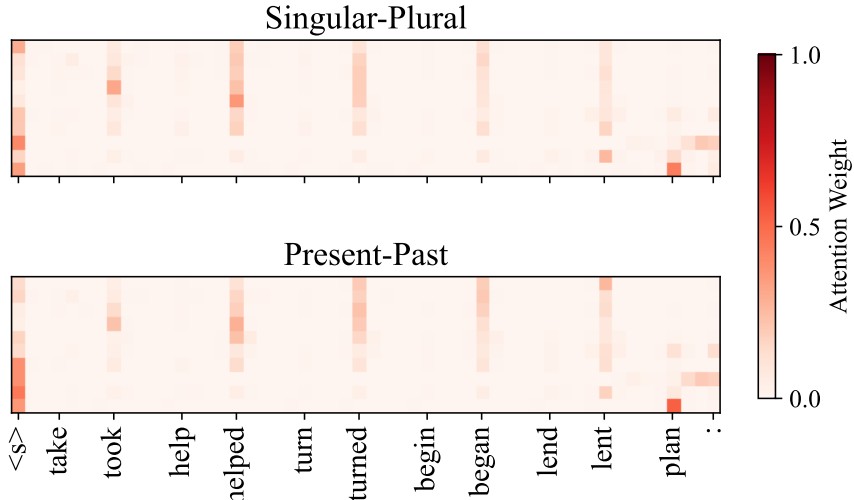

Figure 23: Attention weight visualizations for the singular-plural and present-past tasks for the attention heads with the top 10 average indirect effects in GPT-J. Across tasks, the attention weights are consistently the strongest on the output tokens of each exemplar.

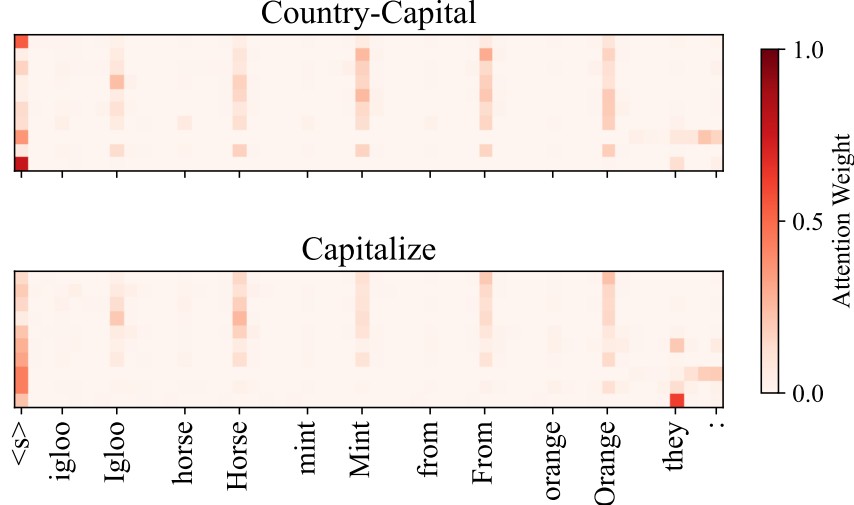

Figure 24: Attention weight visualizations for the country-capital and capitalize tasks for the attention heads with the top 10 average indirect effects in GPT-J. Across tasks, the attention weights are consistently the strongest on the output tokens of each exemplar.

To further investigate whether the heads identified via causal mediation analysis are "induction heads", we compute the prefix-matching score for each head in GPT-J. We follow the same procedure as described in (Olsson et al., 2022; Wang et al., 2022a), which computes the prefix-matching score as the average attention weight on a token $B$ when given a sequence of the form $[A, B, \ldots, A]$. This is

measured on sequences of repeated random tokens. We do this for each head in GPT-J with results shown in Figure 25b.

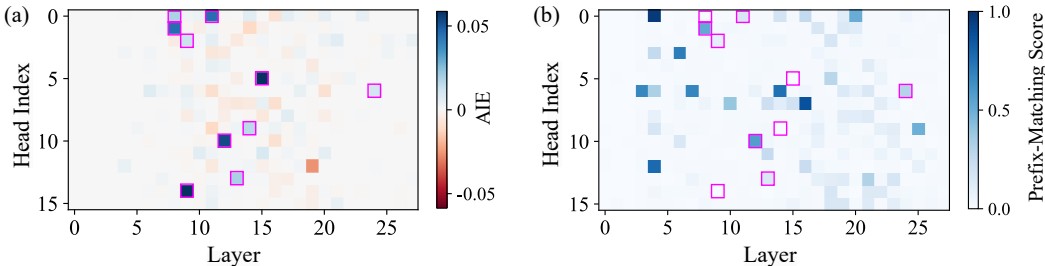

Figure 25: (a) Average Indirect Effect (AIE) per attention head for GPT-J. (b) Prefix-matching score per attention head for GPT-J. For both (a) and (b), we highlight in pink the top 10 heads by AIE. There are three heads that have both a relatively high AIE and prefix-matching score (Layer-Head Index = 8-1, 12-10, and 24-6). There are also several heads with high AIE that do not have a high prefix-matching score, and vice-versa.

We find that three of the heads out of those with the top 10 highest AIEs (Figure 25) also have high prefix-matching scores. In terms of "Layer-Head Index", these are heads 8-1, 12-10, and 24-6, with prefix-matching scores of 0.49, 0.56, and 0.31, respectively.

While (Elhage et al., 2021; Olsson et al., 2022) show that induction heads play a critical role in copying forward previously seen tokens, our results show that they are also among the set of heads, $\mathcal{A}$, that have the highest AIE when resolving few-shot ICL prompts.

There are several other heads we identified with relatively high causal effect that have the same attention pattern activation on few-shot ICL prompts, but do not produce the same "induction" attention pattern on sequences of random repeated tokens.

This suggests that while induction heads play a role in the formation of function vectors, there are other heads that also contribute relevant information that may not be induction heads of the type observed by (Elhage et al., 2021; Olsson et al., 2022).

# I  DECODING VOCABULARY EVALUATION

Table 19: Additional tasks and the top 5 vocabulary tokens of their decoded FV. Across a variety of outputs, most encodings are aligned to the task they were extracted from.

| Task $t$ | Tokens in the distribution $D(v_t)$ in order of decreasing probability |
|---|---|
| Capitalize First Letter | 'CN', 'DR', 'RR', ' Ct', 'Ct' |
| Country-Currency | ' Japanese', ' Chinese', ' Arabic', ' Russian', ' American' |
| English-German | ' âĶĨ', ' ËĨ', ' è', 'actual', ' ç¥l' |
| English-Spanish | ' âĶĨ', ' è', ' ç¥l', ' masc', 'operator' |
| Landmark-Country | ' Germany', ' Japan', ' Netherlands', ' Italy', ' Spain' |
| Lowercase First Letter | 'dr', ' nr', ' lc', ' mc', ' mr' |
| National Parks | ' Connecticut', ' California', ' Wisconsin', ' Netherlands', ' Pennsylvania' |
| Park-Country | ' Netherlands', ' Germany', ' Japan', ' Italy', ' Mexico' |
| Person-Sport | ' basketball', ' football', ' soccer', ' baseball', ' tennis' |
| Product-Company | ' Microsoft', ' Motorola', ' Samsung', ' Disney', ' IBM' |
| Sentiment | ' positive', ' negative', 'positive', 'negative', ' neutral' |
| Synonym | ' edible', ' adjective', ' noun', ' slang', ' caster' |

Here, we present more results on the evaluation of decoding vocabularies of FVs, over additional datasets. Across the tasks, we affirm that output spaces seem to be frequently encoded in the FVs, as can be seen in Table 19. In particular, in cases such as sentiment that follow a rigid pattern, the tokens referring to the output distribution for sentiment is well encoded. On the other hand, some tasks like language translation do not have output spaces well-encoded in the FVs.

## J  SCALING EFFECTS

Can we consistently locate function vectors given various sizes of a single language model architecture? We test this by observing all sizes of Llama 2, ranging from 7B parameters to 70B. We use the same methods as in §3.1, adding function vectors to each layer of the model and observing accuracy on our subset of 6 tasks at each layer.

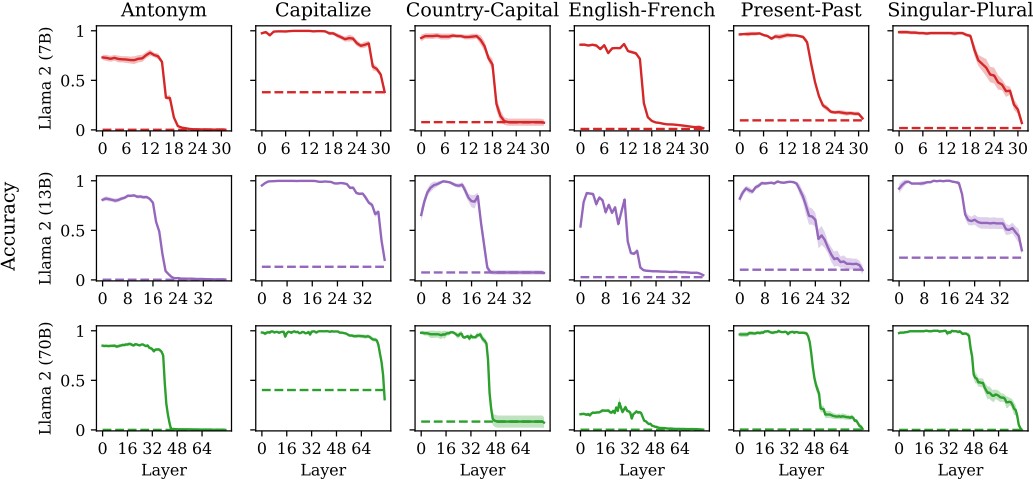

Figure 26: Zero-shot accuracy across Llama 2 model sizes in zero-shot settings. We show accuracies before adding the function vector (dotted lines) and after adding the FV to a specific layer (solid lines).

We find that results (Figure 26) are largely consistent across model sizes. Function vectors generally result in the highest zero-shot accuracies when added to the early to middle layers; this is true regardless of the total number of layers in the model.

# K  COMPOSITION ON OTHER MODELS

In this section we include additional composition results for Llama 2 13B and 70B models in Tables 20 and 21 respectively.

Table 20: The accuracy of ICL, calculated FV $v_{BD}$ zero-shot interventions, and vector-composed $v_{BD}^*$ zero-shot interventions when performing several list-oriented tasks on LLaMA 2 (13B).

| Task | ICL (ten-shot) | $v_{BD}$ (FV on zero-shot) | $v_{BD}^*$ (sum on zero-shot) |
|---|---|---|---|
| Last-Antonym | $0.53 \pm 0.02$ | $0.26 \pm 0.03$ | $0.17 \pm 0.02$ |
| Last-Capitalize | $0.94 \pm 0.01$ | $0.63 \pm 0.03$ | $0.70 \pm 0.03$ |
| Last-Country-Capital | $0.86 \pm 0.02$ | $0.73 \pm 0.02$ | $0.37 \pm 0.04$ |
| Last-English-French | $0.75 \pm 0.02$ | $0.32 \pm 0.02$ | $0.12 \pm 0.02$ |
| Last-Present-Past | $0.96 \pm 0.01$ | $0.22 \pm 0.02$ | $0.24 \pm 0.02$ |
| Last-Singular-Plural | $0.89 \pm 0.01$ | $0.43 \pm 0.03$ | $0.53 \pm 0.03$ |
| Last-Capitalize-First-Letter | $0.85 \pm 0.02$ | $0.89 \pm 0.02$ | $0.89 \pm 0.02$ |
| Last-Product-Company | $0.47 \pm 0.01$ | $0.44 \pm 0.03$ | $0.60 \pm 0.03$ |

Table 21: LLaMA 2 (70B)

| Task | ICL (ten-shot) | $v_{BD}$ (FV on zero-shot) | $v_{BD}^*$ (sum on zero-shot) |
|---|---|---|---|
| Last-Antonym | $0.67 \pm 0.03$ | $0.43 \pm 0.03$ | $0.47 \pm 0.03$ |
| Last-Capitalize | $0.99 \pm 0.00$ | $0.93 \pm 0.01$ | $0.95 \pm 0.01$ |
| Last-Country-Capital | $0.81 \pm 0.03$ | $0.91 \pm 0.02$ | $0.94 \pm 0.02$ |
| Last-English-French | $0.84 \pm 0.02$ | $0.13 \pm 0.01$ | $0.17 \pm 0.03$ |
| Last-Present-Past | $0.98 \pm 0.01$ | $0.93 \pm 0.01$ | $0.94 \pm 0.01$ |
| Last-Singular-Plural | $0.98 \pm 0.01$ | $0.69 \pm 0.04$ | $0.69 \pm 0.04$ |
| Last-Capitalize-First-Letter | $0.67 \pm 0.03$ | $0.60 \pm 0.02$ | $0.68 \pm 0.03$ |
| Last-Product-Company | $0.49 \pm 0.02$ | $0.34 \pm 0.01$ | $0.34 \pm 0.03$ |

## L  EVALUATING FUNCTION VECTORS ON CYCLIC TASKS

In Appendix A, we discuss whether function vectors (FVs) can be thought of as simple word vector offsets, and show that cyclic tasks (such as antonyms) are a counterexample to this claim. In this section we report the causal effects of function vectors on two additional tasks with cyclic subsets —"next-item" and "previous-item" — providing further evidence that function vectors are not just simple semantic vector offsets but instead can be thought of as a trigger of nontrivial functions.

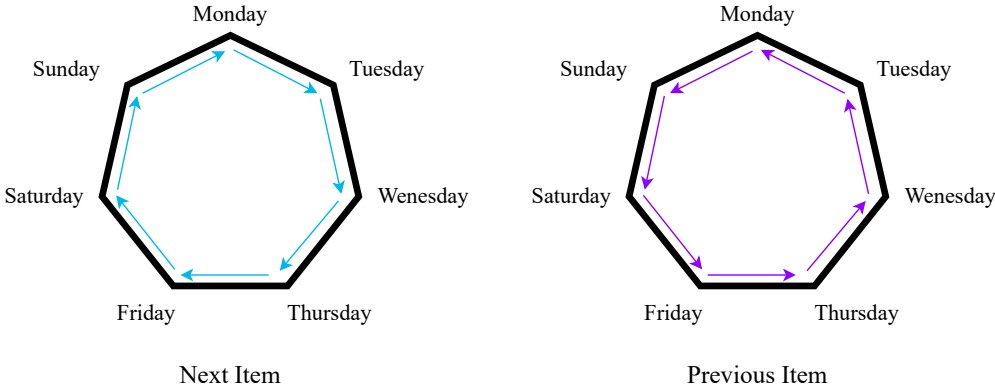

Figure 27: An example of cyclic structure for days of the week, which is a subset of the data for both the Next-Item and Previous-Item tasks. The cycles in each task follow the opposite order (e.g. Next-Item(Monday) = Tuesday, but Previous-Item(Monday) = Sunday.

The "next-item" task contains pairs of words which are related via the abstract idea of "next". The "previous-item" task contains the reciprocal version of the word pairs in the "next-item" task. Flipping the direction in this manner means each pair communicates the idea of "previous" instead. Both tasks are collected over a heterogeneous set of sequential data that includes cyclic types such as days of the week, months of the year, and letters of the alphabet, as well as non-cyclic types such as numbers and roman numerals. We include samples of example data pairs for these datasets in Appendix E.

However, a single ICL example for the Previous-Item task might look like "Q: Friday\nA: Thursday\n\nQ: six\nA: five\n\nQ: a\nA:z\n\nQ: VII\nA:VI\n\nQ: September\nA:". The model would ideally be able to answer "August" given this ICL prompt.

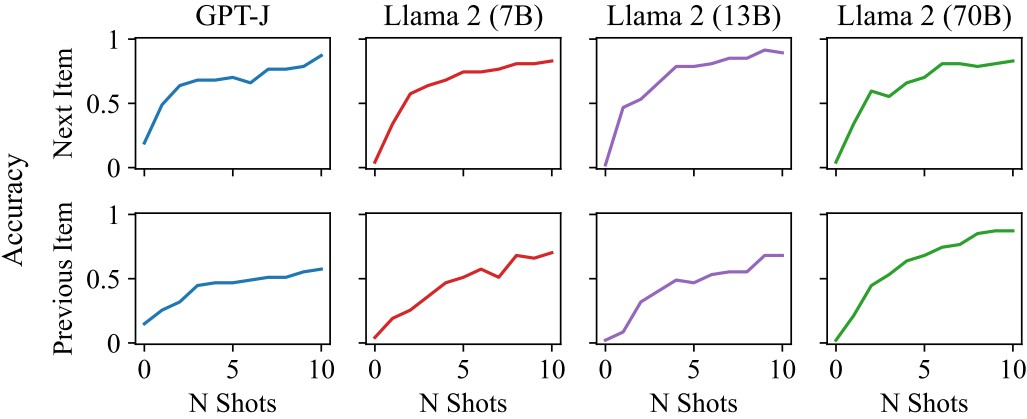

Figure 28: ICL performance on the "Next-Item" and "Previous-Item" cyclic tasks for 4 different models. The performance is usually better for the next-item task than for the previous-item task. However, 10-shot performance suggests these models are able to perform these tasks fairly well.

In Figure 28, we report the ICL n-shot performance of each of these two tasks for GPT-J, and each model in the Llama 2 family. We find that the models perform this task well given 10 example pairs, with the performance of the next-item task being higher than the previous-item task. Performance generally increases when more examples are provided.

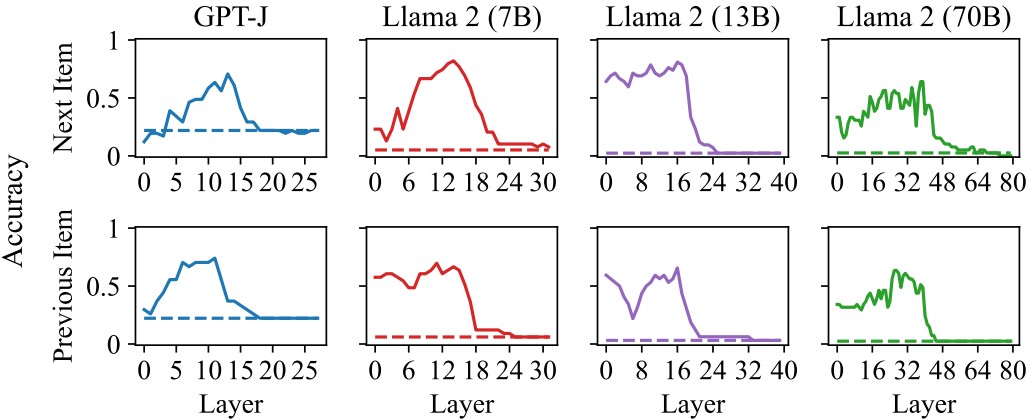

Figure 29: Zero-shot accuracy on the "Next Item" and "Previous Item" cylic tasks for GPT-J and all sizes of Llama 2. We show model accuracies before adding the function vector (dotted lines) and after adding the FV to a specific layer (solid lines). The function vector improves performance significantly for both tasks compared to the model's zero-shot baseline.

We extract a function vector for each of these tasks and evaluate their performance in the zero-shot setting, adding the function vector to different layers of the network. In Figure 29, we report the zero-shot accuracy of each model before adding the function vector with a dashed line, and the accuracy after adding a function vector with a solid line. We see that the function vector significantly improves performance for each task compared to the zero-shot baseline. In addition, the trends for these datasets generally follows those previously reported for other tasks (cyclic or not). The peak performance is achieved when adding the FV to early-middle layers, and there is a sharp drop in performance about $2/3$ of the way through the network.

Table 22: A few example outputs of adding the "next-item" and "previous-item" function vectors to layer 9 of GPT-J. We see that the function vectors are able to correctly trigger the cyclic behavior of "next" or "previous" when presented with a boundary case, while the base model usually defaults to copying the input query.

| Input Prompt: | 'Q: Sunday\nA:' | 'Q: December\nA:' | 'Q: z\nA:' | 'Q: seven\nA:' | 'Q: 21\nA:' | 'Q: Monday\nA:' | 'Q: January\nA:' |
|---|---|---|---|---|---|---|---|
| **GPT-J** | Sunday | December | z | eight | I don't know | Tuesday | January |
| **GPT-J+Next-Item FV** | Monday | January | a | eight | 22 | Tuesday | February |
| **GPT-J+Previous-Item FV** | Saturday | November | y | six | 20 | Sunday | December |

In Table 22, we include a few example outputs of zero-shot prompts for both the baseline model and the model when we add the corresponding function vectors of either "next-item" or "previous-item", showing their ability to correctly induce the "next" or "previous" cyclic behavior.

For antonyms, the cyclic behavior gives a contradiction in two additions. That is, given a word $w_1$ and a vector offset $v$ that can give the antonym of $w_1$, then we expect to return to $w_1$ after adding $v$ again (i.e $w_1 + 2 * v = w_1$). The cyclic subsets in studied here have longer cycles (e.g. for days of the week, and an offset $v'$, we'd expect $w_2 + 7 * v' = w_2$), but the same reasoning applies. Because FVs can trigger the corresponding cyclic behavior, this provides additional evidence that they are not just doing simple word vector arithmetic.

## M    AN ALTERNATIVE EVALUATION OF FUNCTION VECTORS

Recall that we define a function vector ($v_t$) for a particular task ($t$) as the sum of the task-conditioned mean activations ($\bar{a}_{\ell j}^t$) over a small set of attention heads ($\mathcal{A}$) (see equation 5, Section 2.3). Given a function vector created in this manner, we can test its causal effects by adding it to a single hidden state of a model at a particular layer and measuring its ability to trigger a particular task behavior (see Appendix B.2 for more details).

An alternative approach to test whether the outputs of the attention heads in $\mathcal{A}$ can trigger a particular task behavior is to instead add their task-conditioned mean activations ($\bar{a}_{\ell j}^t$) to their corresponding layer's hidden states, and to do so at every layer that is implicated by the heads contained in $\mathcal{A}$. This is in contrast to the FV, which adds all these attention head outputs to a single layer.

As the model performs computation at layer $k$, the alternative approach updates the hidden state $\mathbf{h}_k$ by adding the task-conditioned mean activations of all heads in $\mathcal{A}$ that output to layer $k$. If we represent the attention heads in $\mathcal{A}$ with (layer, head index) tuples, then we write the updated hidden state $\mathbf{h}_k^{'}$ as:

$$\mathbf{h}_k^{'} = \mathbf{h}_k + \sum_{(\ell,j)\in\mathcal{A} \,|\, \ell=k} \bar{a}_{\ell j}^t \tag{16}$$

We perform the update intervention as specified in equation 16 for all layers represented by $\mathcal{A}$.

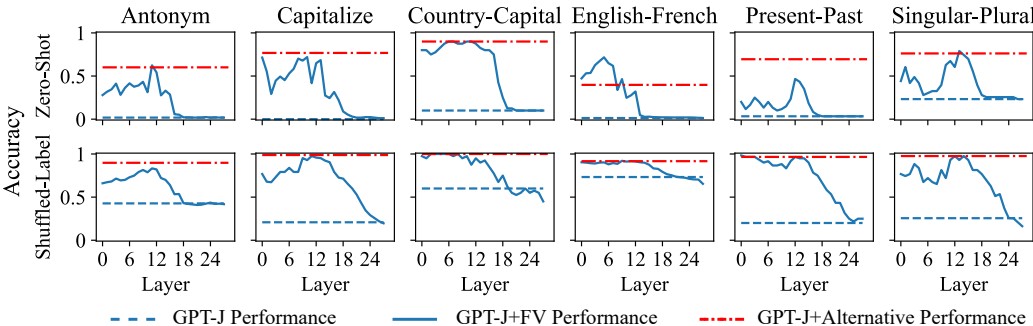

Figure 30: Comparing the causal effects of function vectors (solid blue line) and an alternative approach (red dashed line), which adds the components of an FV to their respective layers instead of at a single concentrated layer. The model baseline in each setting is shown with a dashed blue line. Zero-shot results for 6 tasks are shown in the first row, and shuffled-label results are shown in the second row. In the zero-shot setting, the alternative approach matches FV performance for most tasks. It performs worse for English-French translation and better on the Present-Past task. In the shuffled-label setting, the alternate approach matches FV performance for all tasks.

In Figure 30, we compare the causal effects of the alternative approach described in equation 16 to the original function vector formulation for both zero-shot and shuffled-label contexts across our 6 representative tasks using GPT-J. The base model performance is shown with a dashed blue line, and the solid blue line shows performance when we add the function vector to layer $\ell$. The results of the alternative approach are shown with a red dashed line.

In the zero-shot setting, the alternative approach matches the performance of the function vector for a majority of the tasks. It performs worse on English-French, and better on the Present-Past task. In the shuffled-label setting, the alternative approach matches the causal effects of function vector for all tasks.

On average, we find that using the alternative approach to measure the causal effects of $\mathcal{A}$ compared to the original function vector formulation works about as well, typically achieving the same peak performance. However, the function vector approach does highlight an interesting phenomena of performance dropoff around 2/3 of the way through the network which is not possible to see when using the alternative approach.

## N    Investigating Function Vector Effects in Vocabulary Space

In the main paper (Section 3.2, Table 6), we examine quantitative evidence that the action of a function vector (FV) cannot be explained by simply boosting a set of words directly encoded by the function vector in the style of Dar et al. (2023), see Appendix A for a more detailed discussion. In this section we examine the causal effects of several function vectors in vocabulary space to understand the relationship between the words that are encoded in a function vector and the words that are boosted by the transformer when we intervene with an FV.

Unlike previous analysis, in this section we investigate how adding a function vector ($v_t$) to layer $\ell$ changes the distribution of log probabilities over a set of relevant tokens ($w_i \in W$) compared to the baseline model's response. That is, for a token $w_i$ we compute:

$$\triangle \text{logprob} = \log(f(p^t \mid \mathbf{h}_\ell := \mathbf{h}_\ell + v_t)[w_i]) - \log(f(p^t)[w_i]) \tag{17}$$

We investigate the tokens with the highest increase in log probabilities under FV intervention and include a few examples of the behavior we observe in Table 23. Here we show a few examples of the tokens with the largest $\triangle$logprob for three tasks: Country-Capital, Antonym, English-French.

Table 23: The tokens with the highest increase in $\triangle$logprob for different queries on three tasks - Country-Capital, Antonym, and English-French (shown in black text). For comparison, we present the $\triangle$logprob of the top tokens we get when decoding the FV via $D(v_t)$ (shown in red text directly below). The $\triangle$logprob of the $D(v_t)$-tokens is much lower than the query-specific answers. In general, the tokens promoted the most correspond to likely answers to the specific query, rather than generic tokens in the output space. In the case of 'wolf' for the English-French task, the model answers incorrectly. However, examining $\triangle$logprob indicates several likely answers are still promoted – showing FVs have causal effects that are not adequately captured using top-1 accuracy.

| | Tokens with largest positive $\triangle$logprob under FV intervention |
|---|---|
| **Country-Capital** | |
| South Africa | ' Pret' (+4.7), ' Johannes' (+4.2), ' Dur' (+4.0), ' Cape' (+3.9), ' Kimber' (+3.7) |
| | ' London' (+1.3), ' Moscow' (+1.1), ' Paris', (+1.0), ' Bangkok', (+0.3) ' Madrid' (+0.2) |
| Syria | ' Damascus' (+4.9), ' Tart' (+4.2), ' Raqqa' (+4.1), ' Dam' (+4.0), ' Aleppo' (+3.8) |
| | ' London' (+2.2), ' Moscow' (+2.1), ' Paris', (+2.1), ' Bangkok', (+1.3) ' Madrid' (+2.1) |
| **Antonym** | |
| temporary | ' perpetual' (+4.4), ' definitive' (+4.3),' everlasting' (+4.1), ' permanent' (+3.7) |
| | ' counterpart' (+1.3), ' lesser' (+0.9), ' destroy' (+0.5), ' negate' (+0.4), ' wrong' (-0.8) |
| static | ' evolving' (+4.0), ' flexible' (+3.8), ' polymorph' (+3.7), ' dynamic' (+3.7) |
| | ' counterpart' (+0.4), ' lesser' (-0.3), ' destroy' (0.2), ' negate' (-0.4), ' wrong' (-1.7) |
| **English-French** | |
| wolf | ' lou' (+6.3), ' ours' (+6.2), ' chau' (+5.8), ' dé' (+5.8), ' Lou' (+5.7) |
| | ' âĶÌ' (-1.4), ' masc' (-0.6), ' ç¥l' (-0.9), ' embr' (+2.4), ' è' (+1.6) |
| advertisement | ' ann' (+7.6), ' aff' (+6.8), 'annon' (+6.7), ' ré' (+6.2), ' pub' (+6.14) |
| | ' âĶÌ' (0.2), ' masc' (1.3), ' ç¥l' (-0.2), ' embr' (-0.5), ' è' (+0.6) |

For the country-capital task, the tokens with the highest increase in log probability typically correspond with likely answers to the specific query, rather than answers to the task in general. For example, given the query 'South Africa', the country-capital FV promotes 'Pretoria', in addition to other cities in South Africa. We compare the 5 tokens with the highest overall increase in log probability, and the top 5 tokens we get from $D(v_t)$, which are shown below these in red. We see a substantial difference between the magnitudes of the $\triangle$logprob for these tokens and the tokens that were promoted the most.

We see a similar trend for Antonyms - where the promoted tokens are all reasonably valid antonyms of the query word, rather than just antonyms in general.

For English-French, the query ' wolf' is not answered correctly under intervention, but the correct translation (' loup') is still promoted by the FV when we examine the $\triangle$logprob. Similarly, for the query 'advertisement', the dataset target is 'publicité', but prefix tokens for another valid french translation (' annonce') are also promoted when examining the $\triangle$logprob distribution - ' ann', and 'annon'.

In conclusion, for the tasks we examine we find that function vectors have strong causal effects even when top-1 accuracy metric does not adequately capture this behavior. Furthermore, we find that the causal effects of the FV do not just generically promote words in the output space, but specific words that are plausible answers for each individual query.

