# OpenReview forum: "Function Vectors in Large Language Models"
_ICLR.cc/2024/Conference — ICLR 2024 poster_

### Official Review · Reviewer_baEP · 2023-10-30

**Soundness:** 3 good
**Presentation:** 3 good
**Contribution:** 3 good
**Rating:** 6
**Confidence:** 3

**Summary:**

The authors devise a simple way to extract compact task representations from LLMs. They further investigate how well these task representations generalize and whether these representations can be meaningfully composed to create new task representations (i.e. whether vector-algebraic operations on them are meaningful).

**Strengths:**

- The paper addresses a highly relevant topic.
- Experimental support is sufficient.
- The findings are likely to be of broad interest.
- The paper is clearly written and is a pleasure to read.

**Weaknesses:**

- I feel that the contribution is not sufficiently clearly placing itself in the context of existing work. In particular, the "related work" section seems brief and insufficient.

There is extensive work on distributed, composable and generalizeable task representations (e.g. Lampinen, A. K., & McClelland, J. L. (2020). Transforming task representations to perform novel tasks. Proceedings of the National Academy of Sciences, 117(52), 32970-32981, but see related work also).

I believe that the impact of the present contribution on our understanding of vectorized task representations (both in LLMs and more generally) is not discussed in sufficient depth. It also seems important to broaden the discussion a little, to include not only work on task representations, but also discuss works that looked into the roles of different attention heads in general (see e.g. Clark, K., Khandelwal, U., Levy, O., & Manning, C. D. (2019). What does bert look at? an analysis of bert's attention. arXiv preprint arXiv:1906.04341.). I.e. why they overlooked/were unable to interpret attention heads as encoding specific tasks. Is the proposed probing method superior, or is it because nobody cared to look, etc.

- As a related issue, I think it is crucial for the authors to explain why the results are important. Currently, the authors say "Our study of function vectors differs qualitatively from these previous works: rather than training a model to create function representations, we ask whether a pretrained transformer already contains compact function representations that it uses to invoke task execution". In a way, I feel that this is underselling their work.

It is clear without doing any research that pre-trained LLMs must have some form of task representations (otherwise they won't be able to do zero/few-shot learning). I think that developing a way to extract those representations is valuable, but a more thorough/careful discussion is crucial to highlight why this is important and how it advances existing work.

**Questions:**

I might have missed it, but I wonder if the authors could provide a metric for the "proportion of task representation encoded by FVs". One potential way to visualize that would be to add lines to figure 4 representing best ICL performance, similarly for table 2, a few-shot baseline column would be very helpful. In other words, I do not fully understand how much task information we lose when we move from ICL to FVs.

A minor suggestion on presentation:
I feel that figures 1 and 2 are introduced much too early. They are confusing without proper context, which is given much later.

---

> ### Author Response · Authors · 2023-11-12
> **Response for Reviewer baEP**
>
> Thank you for your thoughtful review. We are glad you found our paper highly relevant and clearly written. We have posted an updated version of the paper including the improvements that you have suggested.  We also discuss several of your suggestions and questions here:
>
> ------
>
> __On the previous work on Lampinen on task representations, as well as Clark on attention heads:__
>
> Our work builds on that previous work and is consistent with the findings from the existing body of work, and it adds some new insights.  We have added this discussion to our updated related work section.
>
> In detail: our work builds upon the attention-weight-based analyses of attention as done by Voita 2018, Clark 2019, Voita 2019, Kovaleva 2019, Reif 2019, Lin 2019, Htut 2019 who found that attention patterns follow many linguistic dependency relationships across a variety of tasks; we also find that attention heads in ICL systematically follow structural dependencies, as shown in Figure 3b. However, that is not the focus of our inquiry: inspired by the observations that attention patterns alone do not fully explain the mechanisms (Jain 2019, Wiegreffe 2019, Kobayashi 2020, Bibal 2022), we focus on another way of understanding the role of attention heads.
>
> The focus of our paper is to ask another question about the specific role of attention heads in ICL contexts: what is the content of the information transported by the attention heads? We pose this question in a way that differs from the probing classifiers proposed in Clark et al. which examine attention maps to characterize attention head behavior for specific linguistic relationships. In contrast we use causal mediation analysis, which does not rely directly on the attention pattern, but on the content of the attention output, and is thus not directly comparable to their approach. Our finding on the transported function-identification information in ICL offers a new window into the nontrivial and interpretable role that attention can play.
>
> ------
>
> __Clarifying the importance of our work.__
>
> Thank you for the suggestions. We agree that the importance of the finding should be clarified, so based on your suggestion we have made several changes to the paper to clarify the framing and importance of the result.
> - We have changed the title of our work to “Function Vectors in Large Language Models”.
> - We have updated the introduction to include a clearer contrast of our work and simple word vector arithmetic, including a detailed discussion in Appendix A.
> - We have also updated our discussion and related works section.
>
> ------
>
> __How much ICL performance do function vectors recover?__
>
> We mention briefly at the beginning of section 3.1 that “we only include those test queries for which the LLM answers correctly given a 10-shot ICL prompt; all accuracy results are reported on this filtered subset.“ Thus, the results presented in figure 4 and table 2 can be understood to be the proportion of the task representation encoded by the FVs, relative to the original model’s ICL performance - which corresponds to 100% for the reported results. However, we agree that we should make this more clear and have added some clarification of the implications of this choice in the revised version of the paper.
>
> Additionally, we have included in Appendix D an analysis where we compare the originally-reported filtered result to the result we get without filtering the test set and include the ICL performance as an oracle baseline for comparison. We find that the performance of our FVs remains very similar under both settings (filtering the test set vs. no filtering)
>
> ------
>
> __Regarding Figures 1 & 2__
>
> Based on your suggestions, we have updated the content of figure 1, and the location of figure 2 to be more in-line with the text describing its setup and results in the updated version of our paper.

---

> ### Comment · Reviewer_baEP · 2023-11-20
> **Thank you**
>
> I have read other reviews and your response. On the one hand, other reviews bring up some valid points/concerns. I especially resonate with R 9YEq's comment on how a deeper look into the FV mechanisms is desirable. On the other hand, I believe that the authors address many of the reviewers' concerns completely or partially.
>
> Overall, therefore, I am leaving my score unchanged.
>
> I would like to encourage the authors to further refine the revised discussion that has to do with previous work (Analyzing the Attention Mechanism section especially). For example, the new sentence "our work draws motivation from the observation that attention patterns alone do not fully explain mechanisms" is a little grammatically confusing (mechanisms of what?). The continuation is also a little confusing "for example Kobayashi et al. (2020) notes the magnitude of transported data is important for insight" - it's not immediately clear how this advances the argument and what is meant by "insight". It would benefit the paper to further rework this paragraph to include a more clear/concrete example of how previous attention analyses failed. I, however, appreciate the ending of that paragraph since it tries to succinctly formulate how their proposed approach allows to gain deeper insight into LLM attention/representations, as compared to previous works.

---

### Official Review · Reviewer_ZAgB · 2023-11-03

**Soundness:** 3 good
**Presentation:** 3 good
**Contribution:** 4 excellent
**Rating:** 6
**Confidence:** 2

**Summary:**

This paper studies the way that tasks are encoded in LLMs, and find that there are a small set of attention heads which can be used to transport the task from in-context learning to unrelated prompts. They refer to these as "function vectors", and they find that these vectors can be used to encourage the model to perform tasks in a zero shot manner. The results support that FVs can be used in a variety of contexts to invoke the task.

**Strengths:**

The paper is a strong contribution towards understanding LLMs in terms of their ability to both encode and respond to ICL task information. The paper is fairly easy to follow: the motivation is clear as LLMs are somewhat of a black box, and I believe this paper has framed their study in the context of something easy to digest (ICL tasks) with clear experimentation. This work has significance as I believe that understanding some of the mechanisms of LLMs and possibly manipulated them for ZS performance will be useful in a number of fields that use LLMs. The approach is fairly creative yet intuitive: I appreciate the way they studied which heads have the most casual influence on tasks, and didn't have any issues understanding the motivations behind how they derive FVs.

**Weaknesses:**

There are a few things I'm a bit unclear about, notably that there is some back and forth on using all heads versus only manipulating the last head (am I understanding that correctly?). Notably, FVs appear to be a vector over tasks: can you describe again how these are applied to the LLM to invoke task behavior? Do you mean that we are only looking at the attention heads to the last token (last column)? Then you sum over these heads for each task and add it to the last hidden state?

Perhaps it would be best to have a diagram, but it seems like some other choices on how to transport the task information to unrelated contexts would have been possible. Could we have not just added the task value (without summing in equation 5) to each attention head leading to computing the next hidden state?

Also, why compute the top attention heads over all tasks and not just have a different distinct set for each task?

**Questions:**

Besides the questions above:

In the intro, I'm not sure a lot of readers will understand the connections to lambda calculus, so I wonder if there's a better way to start of this work.

In 2.2 J is mentioned but not defined: Number of heads?

---

> ### Author Response · Authors · 2023-11-12
> **Response for Reviewer ZAgB**
>
> Thank you for your review, we’re glad you found our paper well-motivated, intuitive, and creative. We have updated our paper submission to answer your questions and incorporate your feedback. We also address your questions here, including pointers to corresponding updates in the paper:
>
> ------
>
> __Clarifying how function vectors are computed and applied to invoke task behavior, and attention heads used__
>
> We have added a section to the appendix (Appendix B) to clarify how attention head outputs are added together to form a function vector, as well as how we add a function vector to the hidden state of a particular layer to invoke task behavior.
> To summarize here - because a function vector is a sum over a set of attention head outputs, adding it to the hidden state is done in the same way that attention outputs are added to the hidden state vector. If the function vector is $v_t$, and the hidden state is $h_l$ we compute the resulting hidden state we get when adding the function vector as $h_l^\{'\} = h_\{l-1\} + m_l + \sum_\{j \leq J\}\{a_\{lj\}\} + v_t$. This can also be written as $h_l^\{'\} = h_\{l\} + v_t$.
>
> You are correct: we focus on the computations being done at the last token of the prompt, so all of these attention heads are located in this token’s stream of predictions.
>
> For our causal analysis we perform a preliminary search over all attention heads (at the final token) to determine their causal effect, but only select a subset of heads to use when creating a function vector. We use multiple tasks to determine this small set of important heads, but we use the activations from only a single task to compute  function vectors. In other words, a function vector is only a vector over a single task.
>
> ------
>
> > Could we have not just added the task value (without summing in equation 5) to each attention head leading to computing the next hidden state?
> >
>
> Yes - This is a good question, of a similar flavor to that of Reviewer 9YEq. We have run this experiment and report the results in Appendix M.
> We find that if we add the attention outputs to its corresponding layer (or at its corresponding ($l, j$) position), we find that this generally matches the performance of our function vector. So the causal effects and results we see are comparable between this alternate formulations.
>
> ------
>
> > Also, why compute the top attention heads over all tasks and not just have a different distinct set for each task?
> >
>
> We found that because there was very substantial overlap in the attention heads that were important for a majority of the individual tasks, having a distinct set for each task performs about the same as using a universal set of heads over all tasks. In Appendix G we include a figure showing the indirect effect of each head, and the overlap of these heads across many tasks.
>
> ------
>
> > In 2.2 J is mentioned but not defined: Number of heads?
> >
> Yes, thanks for pointing this out. We clarify this in the update to the paper.

---

### Official Review · Reviewer_cNFK · 2023-11-08

**Soundness:** 2 fair
**Presentation:** 3 good
**Contribution:** 3 good
**Rating:** 6
**Confidence:** 3

**Summary:**

This paper works on the analysis of transformer language models. It investigates the information of attention heads and identifies a small number of heads to construct the proposed Function Vector that contains the most causal information. To evaluate the ability of the Function Vector, the paper constructs a set of composable ICL tasks and shows that such a vector not only has some semantic abilities like word embeddings but also has some calculation abilities.

**Strengths:**

* The paper does some deep investigation into transformer language models, especially large ones, which can further help us better understand how they work.
* The structure of the paper is clear and is to follow. Figures in the paper are drawn well.
* Experiments are done on different model sizes and model structures. The paper shows great performance improvement of FV compared to Layer Avg.

**Weaknesses:**

* Though the paper mainly investigates attention heads, I’m still wondering why attention heads contain such information rather than the MLP layers. I find some papers [1], and [2] seem to find that MLP is more critical from the information or memory aspect. Can the paper discuss more on this part, which would better explain the proposed findings?

  [1]. Transformer Feed-Forward Layers Build Predictions by Promoting Concepts in the Vocabulary Space

  [2]. MASS-EDITING MEMORY IN A TRANSFORMER

* I think the step from identifying useful attention heads to the proposed FV vector sounds empirical. For example, can the author explain how to determine the number of selected heads? I find that the paper uses |A| = 10 attention heads for GPT-J and scales the number for larger models. However, this point including the scaling and the initial 10 heads for GPT-J needs more explanation. Also, can the paper give more description about directly summing the head outputs together as defined in Eq. (5)? Has the paper compared with other ways like summing them up with different weights?

* For experiments, I find though the paper tested 40 tasks, they are very simple. Thus, can the paper think of some more difficult ones? Meanwhile, I think if the paper can add some analyses about the FV ability under model scaling laws, it can be better.

**Questions:**

Please check the weakness part.

---

> ### Author Response · Authors · 2023-11-12
> **Response to Reviewer cNFK**
>
> Thank you for your review. We’re glad you found our paper clear and easy to follow. We have updated our paper submission to answer your questions and incorporate your feedback. We also address your questions here, including pointers to corresponding updates in the paper:
>
> ------
>
> >Though the paper mainly investigates attention heads, I’m still wondering why attention heads contain such information rather than the MLP layers. … Can the paper discuss more on this part, which would better explain the proposed findings?
> >
> Our work is consistent with previous findings (Meng 2022, Geva 2023) that MLP layers do play a role in enriching hidden states with information stored in the model. Yet both Meng and Geva also found direct evidence that attention heads at the final token transport some important information; in our current work, we choose to investigate the transport of information through the model. The attention mechanism we study is interesting because unlike information storage that utilizes different MLPs for different pieces of information (See Meng 2022 appendix B Figs 10, 14), we find a tiny set of attention heads that serve as common mediators for many very different ICL tasks (Appendix G).
>
> In addition, our research is asking an open question that is not answered by Meng et al: what kind of information is being transported by attention heads? We find that within ICL contexts, attention heads carry a kind of information that was not previously noticed by Meng: we find an encoding of the task itself, rather than a specific answer to a task.
>
> We have added an extended discussion in Appendix A comparing and contrasting our results to the previous "vocabulary space" findings of Geva 2022.
>
> ------
>
> > For example, can the author explain how to determine the number of selected heads?
> >
> For GPT-J we tested using 1-30 attention heads to construct function vectors and found that the increase in performance for many tasks begins to plateau when using about |A| = 10 attention heads.
>
> For the larger models, the number of attention heads is much larger, but we maintain about the same ratio of ~2% of the attention heads. The total number of attention heads for each model is: GPT-J (6B) = 448, LlaMa 2 (7B)  = 1024, LlaMa 2 (13B) = 1600, GPT-NeoX (20B) = 2816, LlaMa 2 (70B) = 5120. For these models we use 10, 20, 50, 50, and 100 heads respectively, corresponding to roughly 2.23%, 1.95%, 3.12%, 1.78%, and 1.95% of the total heads.
>
> We have added information to Appendix C with a figure showing these experiments and additional discussion of this choice of the number of selected heads.
>
> ------
>
> __Clarifying the creation of function vectors in equation 5__
>
> We have added an appendix section (Appendix B) to clarify how attention head outputs are added together to form a function vector and how we add a function vector to the hidden state of a particular layer.
>
> Because the function vector is a sum of average contributions of attention heads, these attention head outputs are already naturally weighted by the transformer. We did perform some preliminary investigations into weighting the function vector when adding it to a particular layer, finding that a weight of 1 was close to optimal. Thus, weighting seemed to not lend any advantage. Additionally, not weighting the FV allows us to keep the function vector as close to the original hidden state distribution as possible.
>
> ------
>
> > For experiments, I find though the paper tested 40 tasks, they are very simple. Thus, can the paper think of some more difficult ones?
> >
> While we do not investigate these tasks extensively in the main paper due to space constraints, we do include a few standard challenging NLP benchmark tasks among the set of tasks we test. For example, in the appendix we show results for a sentiment analysis task (SST2), an entity recognition task (CoNLL2003), a question-answering task (commonsenseQA), and a text classification task (AG News).
>
> These can be found in Appendix E.
>
> In addition, reviewer 9YEq has suggested analyzing tasks with cyclic structure that cannot be done with a simple vector offset, and we have created two new datasets of this flavor. We present results on these datasets in Appendix L.
>
> ------
>
> >Meanwhile, I think if the paper can add some analyses about the FV ability under model scaling laws, it can be better.
> >
> We agree, thank you for this suggestion.
>
> In Appendix J in our updated submission we show how FVs perform for different model sizes of the same model family (Llama 2). In general, the ability of FVs to trigger task execution seems to improve with scale. In addition, we have added to the appendix (Appendix E.3) additional results for many individual tasks for Llama 2 (13B) and Llama 2 (70B) for a deeper analysis of these scaling effects.

---

> > ### Author Response · Authors · 2023-11-20
> >
> > As we have not yet received a response, we would like to remind Reviewer cNFK that we have updated our paper submission to answer your questions. Specifically, appendix sections B, C, and E were all added or expanded in response to your suggestions and questions, and we feel they have strengthened the paper. We invite the reviewer to examine these updates and consider increasing your score if your questions have been addressed.

---

> > > ### Comment · Reviewer_cNFK · 2023-11-20
> > > **post-rebuttal**
> > >
> > > Thank you for your detailed reply. I read the review and most of my concerns have been addressed. I'd like to increase my score to 6.

---

### Official Review · Reviewer_9YEq · 2023-11-09

**Soundness:** 3 good
**Presentation:** 3 good
**Contribution:** 2 fair
**Rating:** 6
**Confidence:** 4

**Summary:**

The paper introduces the idea of function vectors, which appear to encode information about a task or transformation to be performed on a word to produce a response word under the task.  Function vectors are obtained by summing the averaged outputs of attention heads at the position of a test task input for heads that individually play relatively large roles across a set of tasks, where the average is obtained from several 10-shot prompts derived from examples of performance of the task.  Even though the attention vectors come from heads sprinkled across layers of a transformer, the resulting function vector can induce an increased tendency to give a task appropriate response when added to the hidden state of a transformer at many of the early layers of the transformer.  Experiments demonstrate robustness of the effects of FV's; show that FV's can sometimes be composed to cause a model to perform a composite task; and provide some evidence that the FV's are doing more than simply specifying the space of possible responses that are consistent with the task.

**Strengths:**

The paper provides an example of an way to probe transformer function that adds to the toolkit for causal analysis of how transformers perform tasks.

The findings are suggestive of aspects of the way in which in context examples give rise to task performance.  In particular, it suggests that a relatively small number of attention head play a relatively crucial role in inducing a transformer to produce any one of many different transformation tasks.  It is also interesting that the FV can be added into just one layer out of ~30-80 layers of a deep transformer and have a strong effect on its tendency to perform the given task.  It seems to have implications for how we understand the state-to-state transitions across layers in the network -- ie that they are in someways roughly interchangeable at least across many layers.

The paper uses a diversity of tasks, model types, and base prompts to assess the effects of FV's.  Despite considerable variability in exact magnitudes of effects, the basic effect holds up across all of these manipulations.

**Weaknesses:**

The paper is relatively phenomenological rather than mechanistic; we see that there effects, but there is relatively little analysis of how they actually occur.

More specifically, I thought that the mechanism of action of the FV's could be more forcefully addressed.  At least for some tasks, it is possible that the FV's work by simply specifying the set of responses consistent with the task. E.g. Country-Capital a sufficient mechanism would be the following set-intersection account:  An input country name (France) specifies a set of words related to France; the FV specifies the set of country names; and the correct response is simply their intersection. I recognize that this would not work for all tasks, but it would word for many.  The paper could be strengthened by a detailed analysis of a selected instances of both kinds of tasks.  Some of the tasks specifically appear to require that the FV, if effective, specifies not just how to transform the input, but how to select a response from a string of preceding elements.  None of the 6 tasks selected were of this type, but there were such tasks in the full set.  Highlighting and analyzing these would strengthen the paper. I discuss this further in some suggestions below.

There were also some details that I would like greater clarity about.  I list these in questions.

**Questions:**

Questions:

The text says 'We can then test the causal effect of an FV by adding it to hidden states at any layer ℓ as the model resolves a prompt and measuring its performance in executing the task.' I don't understand exactly how the vector is being 'added to hidden states at any layer'. Am I right that this vector is being added to the final state vector at the top of the lth transformer block?  How is this done given that this state vector typically has dimension equal to the number of heads times the dimensionality of a single head?

How was the 'direct decoding' of FV's performed?  What is the 'optimization' used 'to reconstruct a ˆvtk that matches the distribution Qtk when decoded'?

How was the approach used to determining the FV selected?  It would be informative to understand alternatives you might have tried.  What happens if the FV is the SET of averaged a_lj, with each inserted into the appropriate l,j position?  If this isn't better than 'adding the LV to the hidden state at layer l', why not?

Suggestions:

It seems to me that a stronger analysis could arise from an examination of how the FV changes the distribution of logpobs of relevant words relative to the baseline prompt, and how this varies as the FV is inserted at different levels.  This approach could be used both for tasks where the set-intersection account could be sufficient, and for tasks where it cannot.

The paper would be strengthened if the authors could provide some results from carefully constructed test sets in which the set of outputs is the same set as the set of inputs (e.g matched antonym and synonym pairs, months, days of week, letters of the alphabet, progression of planets, chemicals in the periodic table) afford possibilities.

It might be even more useful to understand tasks in which the task response depends strictly on the words in the context, and we are assured that these words did not enter in to the construction of the FV.  For example, a set of n10-shot prompts involving a random subset chosen from a larger set of words could be used to construct FVs for the m 'chose pth word from list of length m' tasks. Then a test set of word lists of length m could be formed from a sample from the remaining words in the larger word set, and these could be factorially combined with the FV's for each of the values of p in m.  Clearly in this case, the FV's cannot themselves activate the set of answer responses -- if anything they'd activate words in the lists used to form the FV's.  So they must be *inducing* later heads to attend to the correct p in the given list in this case.

Constructing test sets in which x and y come from the same set of words could be helpful.  Antonym and synonym tasks could address this if so constructed, but little detail was provided about the sets of x and y items used, and I did not see a description of explicitly reversed stimulus-response pairs.  (It is possible, for example, that the antonyms tended to be the negative member or less advantaged member of a pair, as the list of decoded words suggests).  I recognize that the full set of tasks include some that make this possibility unlikely (specifically, the chose-item-in-a-specific-list position), since (although not described) it seems likely that x is something uninformative like 'answer'.

---

> ### Author Response · Authors · 2023-11-12
> **Response to Reviewer 9YEq**
>
> Thank you for your review and helpful suggestions. We have updated our paper submission to answer your questions and incorporate your feedback. We also address your questions here, with pointers to the corresponding updates in the paper:
>
> ---
>
> >The paper is relatively phenomenological rather than mechanistic; we see that there effects, but there is relatively little analysis of how they actually occur.
> >
> In our paper we focus on how LLMs trigger task execution via a functional mechanism. We test the causal effects of states under interventions as we add a function vector, so we’re not just looking at correlations. Furthermore, the tests are done under dramatic domain shifts, which is evidence of very strong generalizable mechanistic effects. We’ve rewritten the discussion in the paper to clarify this contribution.
>
> ---
>
> __Clarifying how function vectors are created and added to a hidden state.__
>
> We have added an appendix section (appendix B) to clarify how attention head outputs are added together to form a function vector and how we add a function vector to the hidden state of a particular layer.
>
> In short, the attention head outputs can be viewed as having the same dimension as the hidden states. Because a function vector is a sum over a set of attention head outputs, adding it to the hidden state is done in the same way that attention outputs are added to the hidden state vector.
>
> ---
>
> >How was the ‘direct decoding’ of FV’s performed?
> >
> To decode a function vector, we use the transformer’s decoder (denoted as $D$ in the paper). Formally, this means that we pass the function vector through the model’s final layer norm and subsequent unembedding matrix to turn it into a distribution over the model’s vocabulary space, similar to how the output of the final transformer layer is decoded to vocabulary space via the same decoder $D$. We would write this as $D(v_t) =$ Unembedding(LayerNorm($v_t$)), where $v_t$ is a function vector.
>
> ---
>
> >What is the ‘optimization’ used to reconstruct a $v_\{tk\}$ that matches the distribution $Q_\{tk\}$ when decoded?
> >
> The optimization process we use to reconstruct a $v_\{tk\}$ starts with a randomly initialized vector $v \in \mathbb{R}^d$. We then minimize the Cross-Entropy Loss between the decoded vocabulary distribution of the random vector, $D(v)$, and our resampled top-k distribution $Q_\{tk\}$ using gradient descent w/ the Adam optimizer. $Q_\{tk\}$ is a distribution containing the top $k$ tokens with the highest probability from $Q_t=D(v_t)$; the rest of the vocabulary is zeroed out. In equation 6 we use the term “CE” but realized that we never pointed out this was Cross-Entropy Loss. We have clarified this in the updated version of the paper.
>
> ---
>
> > What happens if the FV is the SET of averaged $a_\{lj\}$, with each inserted into the appropriate $l,j$ position? If this isn’t better than ‘adding the FV to the hidden state at layer $l$’, why not?
> >
> Great question. We have added this analysis to appendix M in the updated submission of the paper.  A function vector can be thought of as adding several attention head outputs to a single layer’s hidden state. But, if we instead add the attention outputs to its corresponding layer (or at its corresponding ($l,j$) position), we find that this generally matches the performance of our function vector.
>
> ---
>
> > The paper would be strengthened if the authors could provide some results from carefully constructed test sets in which the set of outputs is the same set as the set of inputs (e.g matched antonym and synonym pairs, months, days of week, letters of the alphabet, progression of planets, chemicals in the periodic table) afford possibilities.
> >
> This is an excellent suggestion. We had similar thoughts with regards to tasks that have a cyclic nature, where the input and output spaces are similar—for example, antonyms and some of these others you have mentioned which cannot be done via a simple vector offset like perhaps some other tasks can.  We have added some discussion about this idea in an updated version of the paper, in addition to an appendix section (appendix A) discussing this in more detail. We also discuss how this contributes to the idea that FVs are not just simple word vector offsets.
>
> We also created two new tasks - (1) “next item” which includes pairs such as (monday, tuesday), (three, four), (g, h), (February, March) etc. and (2) “previous item” which has similar pairs, but in the other direction: (tuesday, monday), (four, three), (h,g), (January, December). We find that function vectors are able to trigger this type of cyclic behavior when the input and output spaces are similar. We have added our analysis of function vectors in this setting to Appendix L in the updated version of the paper.
>
> ---
> __Examining function vectors in vocabulary space__
>
> We analyze this in Appendix N, and we find that function vectors have causal effects beyond what is captured by top-1 accuracy.

---

> > ### Author Response · Authors · 2023-11-20
> >
> > As we have not yet received a response, we would like to remind Reviewer 9YEq that we have updated our paper submission to answer your questions. Specifically, appendix sections A, B, L, M, and N were all added in response to your suggestions and questions, and we feel they have strengthened the paper. We invite the reviewer to examine these updates and consider increasing your score if your questions have been addressed.

---

> > ### Comment · Reviewer_9YEq · 2023-11-21
> > **Paper improved, shifting rating into accept range**
> >
> > I see that the paper has improved, and I'm planning on shifting my rating into the acceptable range.  Some minor improvements and further questions whose answers could effect my final level of enthusiasm are below.
> >
> > First and most simply, please be explicit in the text with a phrase that a_lj is not the head output as in Vaswani et al but its projection through a subsequent matrix that scales it up to the model dimension, then refer to the appendix for the details.
> >
> > Second, I'm unclear about the new cyclic tasks.  The authors say 'We find that function vectors are able to trigger this type of cyclic behavior when the input and output spaces are similar.'  What does this mean?  Perhaps relatedly, are the shorts always from the same domain (e.g. Months) and is the domain of the probe item also from that same domain?  Please clarify exactly what was done here.
> >
> > The authors pushed back (politely, I'm not complaining) on my reaction that the paper was more phenomenological than mechanistic.  These are slippery terms, as is the term 'causal', which is often confused with mechanism.  They argue that they have demonstrated causality, and I agree, but that's not the same as mechanism.  We could know that red wine causes headaches without knowing the mechanism -- indeed an article in today's NYT describes research that suggests a possible mechanism, addressing a longstanding mystery about a known causal relationship.  In general a mechanism describes the intervening (often unobserved) processes that mediate between cause and effect.  I still think these is little of this here.  There's some discussion or some mechanism-like ideas in the extensive appendix, but I think the paper would be strengthened if the 'Discussion' could say that there is a lot more work that needs to be done to fully understand exactly how FV's have their causal effects.
> >
> > I made a suggestion that related to this which (unless I missed it) the authors did not respond to:
> >
> > *It might be even more useful to understand tasks in which the task response depends strictly on the words in the context, and we are assured that these words did not enter into the construction of the FV. For example, a set of n10-shot prompts involving a random subset chosen from a larger set of words could be used to construct FVs for the m 'chose pth word from list of length m' tasks. Then a test set of word lists of length m could be formed from a sample from the remaining words in the larger word set, and these could be factorially combined with the FV's for each of the values of p in m. Clearly in this case, the FV's cannot themselves activate the set of answer responses -- if anything they'd activate words in the lists used to form the FV's. So they must be inducing later heads to attend to the correct p in the given list in this case.*
> >
> > Time is short for further experiments on this, but perhaps there is time to comment on the fact that the authors do include versions of tasks like those suggested here (first and last from a list, for example).  At least in these tasks, the FV derived from earlier lists seems likely to be instructing later layers of the transformer to attend to particular positions in the immediately preceedig list.  If so, this could be discussed in the section on vector algebra, and would move the paper a little further toward addressing mechanism.
> >
> > Looking forward to further reactions, will look in again after 9 pm Tuesday PDT

---

> ### Author Response · Authors · 2023-11-21
>
> Thank you for your interest in the paper, and your helpful feedback! We address your additional comments below:
>
> ---
>
> >please be explicit in the text with a phrase that a_lj is not the head output as in Vaswani et al but its projection through a subsequent matrix that scales it up to the model dimension, then refer to the appendix for the details.
> >
> We have updated the formulation in section 2.2 to make this explicit.
>
> ---
>
> __Additional Cyclic Tasks__
>
> The cyclic tasks are two additional tasks. Next Item and Previous Item. Both tasks are collected over a heterogeneous set of sequential data that includes cyclic types such as days of the week, months of the year, and letters of the alphabet, as well as non-cyclic types such as numbers and roman numerals. For example, a single ICL example for the Previous-Item task might look like “Q: six\nA: five\n\nQ: a\nA:z\n\nQ: VII\nA:VI\n\nQ: September\nA:”. The model would ideally be able to answer “August” given this ICL prompt.
>
> We collect function vectors from ICL examples such as the one above to get a “Previous-Item” function vector. We then test it on zero-shot examples such as “Q: Monday\nA:”, which when we add the function vector, the model outputs “Sunday”. And for similar queries we observe the following behavior: Monday+FV = Sunday, Sunday+FV=Saturday, Saturday+FV=Friday, Friday+FV=Thursday, Thursday+FV=Wednesday, Wednesday+FV=Tuesday, Tuesday+FV=Monday.
>
> We call the input and output spaces similar because the task defines a function that maps numbers to numbers, days of weeks to days of weeks, and months to months. In addition, we say the function has cyclic structure because if the function is applied repeatedly it ends up arriving at the same data point for certain subsets. For example, Monday will map back to Monday again. This cyclic structure supports the arguments in Appendix A (point 1), and at the bottom of page 48.
>
> ---
>
> __Causality and Mechanisms__
>
> We appreciate the reviewer’s clarity around the distinction between causality and full explication of mechanisms. Our paper has taken one step in elucidating a mediating mechanism of ICL: function vectors. But we agree that we have not characterized the inner workings of these function vectors themselves, which we leave for future work.  We have updated the Discussion section to reflect this.
>
> As described in our updated discussion, prior to this work the mechanisms underlying ICL were largely unknown. Our identification of function vectors can be thought of as analogously moving from knowing “going to a party causes headaches” to “red wine causes headaches”. Although we haven’t elucidated every detail of how language models perform ICL, the identification of function vectors as a mediator is a key step forward and opens the path for follow-up work.
>
> ---
>
> __Tasks that include choosing a word from a list__
>
> Apologies for overlooking this comment. We have updated section 3.3 to include some discussion about the difference between tasks that require "word-selection" and "word-transformation" and potential implications for future research.

---

> > ### Comment · Reviewer_9YEq · 2023-11-22
> > **Thanks, final comment**
> >
> > Thanks for engaging with my comments and questions.  I think the paper would benefit from including some of the content of the following paragraph in the relevant section of the supplement:  I thought the explicit inclusion of the example removed my lingering uncertainty. "The cyclic tasks are two additional tasks. Next Item and Previous Item. Both tasks are collected over a heterogeneous set of sequential data that includes cyclic types such as days of the week, months of the year, and letters of the alphabet, as well as non-cyclic types such as numbers and roman numerals. For example, a single ICL example for the Previous-Item task might look like “Q: six\nA: five\n\nQ: a\nA:z\n\nQ: VII\nA:VI\n\nQ: September\nA:”. The model would ideally be able to answer “August” given this ICL prompt."
> >
> >   I agree with the other reviewers that the best rating for this paper is a 6.  I hope there is room for this paper in the program.  Like others I encourage harder tasks, and further exploration of how the FVs result in successful performance, as directions for future work.

---

> > > ### Author Response · Authors · 2023-11-22
> > >
> > > We are very grateful for your interest and support of our work. Your feedback has been valuable in strengthening the clarity of the paper. We have updated Appendix L to include this ICL example and to be explicit about the data used to construct each dataset.
> > >
> > > We note the form does allow you to change your official score and we would appreciate it if you would consider doing so that it’s more visible to the AC, thank you!

---

> > > > ### Comment · Reviewer_9YEq · 2023-11-22
> > > > **I have changed my official score in my original review to 6**
> > > >
> > > > In view of extensive clarifications and further revisions, I have edited my original review, changing my official score to 6.  I believe the paper helps advance the understanding of LLMs.

---

### Author Response · Authors · 2023-11-12

We would like to thank each of the reviewers for their thoughtful and constructive feedback. We have updated our paper submission to address reviewer comments so far. We provide a summary of key improvements to the paper we have made based on reviewer comments, and we welcome further comments, questions or suggestions. We respond to each individual reviewer in separate responses.

--------

- We have added several sections to the appendix with additional results related to comments and questions from individual reviewers.
- We have updated the introduction, including figure 1, to provide a better framing of the importance of our work.
- We include a discussion of the contrast between our work and previous word vector work, including a more detailed discussion in Appendix A.
- We have updated our related work section and discussion to clarify the context and contribution of the work.
- We have updated the title of our paper to be “Function Vectors in Large Language Models” to clarify the focus of the work.

--------

__Clarification on the construction of a function vector, and adding it to a hidden state (Reviewer 9YEq, cNFK, ZAgB)__

Several reviewers have asked for clarification on how we construct a function vector, and how it is added to a transformer hidden state. We have added an appendix section (Appendix B) to clarify how attention head outputs are added together to form a function vector, and how a function vector is added to a hidden state.

To summarize, attention head outputs can be viewed as having the same dimension as transformer hidden states. Because a function vector is a sum over a set of attention head outputs, adding it to the hidden state is done in the same way that attention outputs are added to the hidden state vector.

--------

__Alternative formulation of function vectors (Reviewer 9YEq, ZAgB)__

While we construct function vectors by adding attention head outputs together into a single vector, it is also possible to either add or replace these outputs at their respective positions in the network (at the corresponding attention head). We have run this experiment report the results compared to our original function vector formulation in Appendix M.

---

### Author Response · Authors · 2023-11-17

We have completed all requested experiments and have posted an updated version of the paper that incorporates all reviewers’ feedback.  We would like to thank the reviewers for their valuable insights.

We invite the reviewers to assess the submission and consider increasing scores if your questions have been addressed.

An overview of the updates in the paper:

- The Discussion has been revised, and the title has been simplified to “Function Vectors in Large Language Models”, in response to reviewer baEP’s request that we clarify why the paper is important.
- Related Work includes more citations and detailed discussion as suggested by baEP.
- Appendix A contrasts function vectors to semantic vector arithmetic, as asked about in different ways by 9YEq, baEP.
- Appendix B contains an explicit definition of the residual stream formulation of attention that we use, which was a point of confusion from reviewers 9YEq, cNFK, ZAgB.
- Appendix C includes data on different sizes of $|A|$ and explains why $|A| = 10$ is used for the main experiments, as requested by cNFK.
- Appendix D measures our results against unfiltered test sets in order to allow comparison to ICL accuracy, as requested by baEP.
- Appendix E includes scaling results, up to Llama 2 (13B) and Llama 2 (70B) as requested by cNFK.
- Appendix L investigates function vectors on additional cyclic tasks where input and output domains are equal, as suggested by 9YEq.
- Appendix M investigates an alternative approach of applying function vectors at many layers instead of one, as suggested by 9YEq, ZAgB, and finds it has similar causal effects to our main results.
- Appendix N investigates fine-grained vocabulary-space action of FVs as requested by 9YEq.
- The main paper has been updated to refer to and discuss the Appendix content.

For each reviewer, we have edited our specific responses with detailed information relevant to their review.

---

### Meta-Review · Area_Chair_prE7 · 2023-12-12

**Metareview:**

### Summary
This paper delves into the concept of function vectors (FVs) within autoregressive transformer language models (LLMs). FVs encapsulate task-specific information and exhibit robustness across various contexts, enabling the execution of tasks even in dissimilar settings. The study uncovers that FVs don't directly execute tasks but trigger the model to perform them through complex computations. While FVs contain task-related data, they cannot be entirely reconstructed from this information alone. Ultimately, this research suggests that LLMs harbor internal representations of versatile functions usable across diverse contexts.

### Decision
The paper is well-written and studies transformers within autoregressive language model modelling setting. The results and the findings of the paper are interesting. The reviews are positive in general about the paper. The authors have done a tremendous job during the rebuttal phase addressing the concerns raised by the reviewers. As it stands now, I think ICLR community would benefit from the insights shared in this paper on transformer language models.

**Justification For Why Not Higher Score:**

The paper is interesting, however, the analysis and the results presented in the paper are limited.

**Justification For Why Not Lower Score:**

The results are interesting and reviews were positive.

---

### Decision · Program_Chairs · 2024-01-16

Accept (poster)